# RoboBench: A Comprehensive Evaluation Benchmark for Multimodal Large Language Models as Embodied Brain

## Abstract

Building robots that can perceive, reason, and act in dynamic, unstructured environments remains a core challenge. Recent embodied systems often adopt a dual-system paradigm, where System 2 handles high-level reasoning while System 1 executes low-level control. In this work, we refer to System 2 as the *embodied brain*, emphasizing its role as the cognitive core for reasoning and decision-making in manipulation tasks. Given this role, systematic evaluation of the embodied brain is essential for advancing robotic intelligence. Yet existing benchmarks emphasize execution success, or when targeting high-level reasoning, suffer from incomplete dimensions and limited task realism, offering only a partial picture of cognitive capability. To bridge this gap, we introduce RoboBench, a benchmark that systematically evaluates multimodal large language models (MLLMs) as embodied brains. Motivated by the critical roles across the full manipulation pipeline, RoboBench defines five dimensions—instruction comprehension, perception reasoning, generalized planning, affordance prediction, and failure analysis—spanning 15 capabilities, 25 tasks, and 6077 QA pairs. To ensure realism, we curate datasets across diverse embodiments, objects with rich attributes, multi-view scenes, and memory-driven navigation, drawing from large-scale real robotic data and in-house collection. For planning, RoboBench introduces a MLLM-as-world-simulator evaluation framework that goes beyond symbolic matching to embodied feasibility, by simulating whether predicted plans can achieve critical object state changes under physical and visual constraints, enabling faithful assessment of long-horizon reasoning. Experiments on 14 state-of-the-art MLLMs reveal fundamental limitations: difficulties with implicit instruction comprehension, spatiotemporal reasoning, cross-scenario planning, fine-grained affordance understanding, and execution failure diagnosis. RoboBench provides a comprehensive scaffold to quantify high-level cognition, clarify the role of the embodied brain, and guide the development of next-generation MLLMs toward more robust robotic intelligence.

## 1 Introduction

Manipulation in dynamic, unstructured environments remains a core challenge for building generalizable embodied intelligence Sun et al. (2024); Clark (1998); Liu et al. (2024c). Such tasks demand not only precise motor control, but also high-level cognition: understanding instructions, perceiving over surroundings, formulating long-horizon plans, inferring affordances, and reflecting on failures Chen et al. (2025b); Zhang et al. (2024). In this scenario, Multimodal large language models (MLLMs) have shown strong potential for these roles, thanks to their strengths in instruction-following, commonsense reasoning, and general planning Li et al. (2024a); Wang et al. (2024). To leverage these capabilities, recent embodied systems integrate MLLMs through a dual-system design Driess et al. (2023); Black et al. (2024); Bjorck et al. (2025), where System 2 performs high-level reasoning and System 1 handles low-level control Liu et al. (2025); Chen et al. (2025a). In vision–language–action (VLA) models, MLLMs are fine-tuned as backbones Black et al. (2024), while in multi-agent frameworks, they serve as high-level planners guiding specialized executors Driess et al. (2023); Huang et al. (2023; 2024); Yuheng Ji (2025). In this work, we refer to System 2 as the *embodied brain*, emphasizing its role as the cognitive core for reasoning and decision-making.

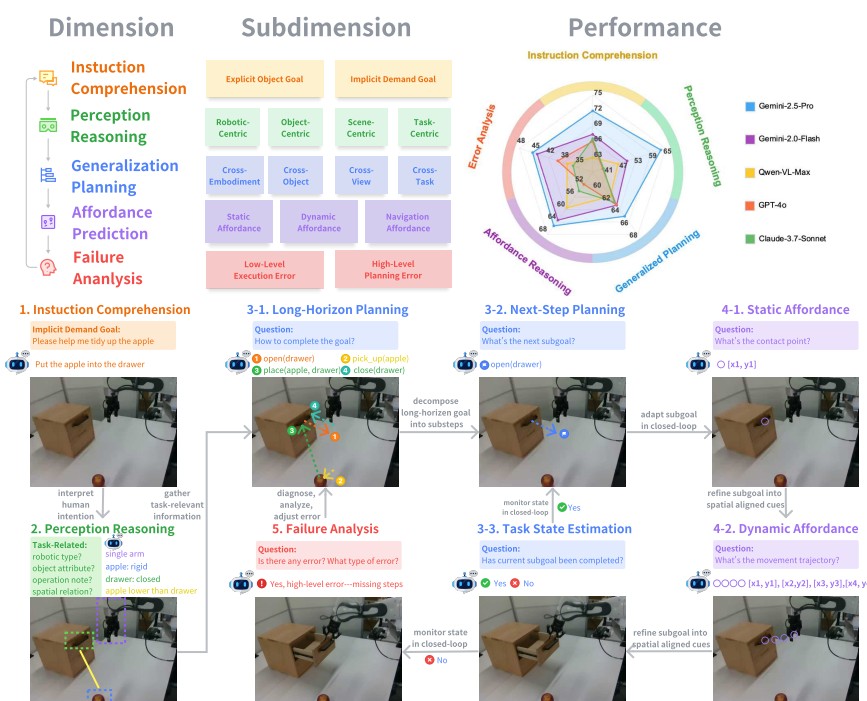

Figure 1: **Overview of RoboBench** We evaluates MLLMs as embodied brains across 5 dimensions, 15 subdimensions, and 25 tasks, with tasks color-coded by type (top left). These dimensions follow the embodied execution pipeline (bottom)—from understanding intent, perceiving the environment, planning and adapting actions, refining subgoals via affordances, diagnosing failures—capturing the core cognitive roles of System 2. Performance comparison (top right) highlights significant gaps among state-of-the-art MLLMs, with Gemini-2.5-Pro achieving the best results.

Given this design, systematic evaluation of the embodied brain is essential, yet current benchmarks remain inadequate. Most existing efforts focus narrowly on overall task success, offering little insight into underlying reasoning James et al. (2020); Liu et al. (2023a); Chen et al. (2025c); Zhang et al. (2024). Even benchmarks that explicitly target embodied cognition exhibit three major shortcomings: (1) fragmented coverage of cognitive abilities, typically isolating perception Majumdar et al. (2024), planning Sermanet et al. (2024); Chen et al. (2024); Li et al. (2024e), or error reflection Liu et al. (2023b); Duan et al. (2024) rather than assessing them as an integrated whole; (2) limited task realism and complexity, relying heavily on simplified simulations Shridhar et al. (2020); Yang et al. (2025); Cheng et al. (2025), or ignoring practical challenges such as diverse embodiment, object properties, and occlusion; and (3) simplistic planning evaluation, often reduced to multiple-choice Chen et al. (2024); Azzolini et al. (2025); Team et al. (2025) or text similarity metrics like BLEU Sermanet et al. (2024) or generic LLM scoring Chi et al. (2024), failing to capture embodied priors such as skill–object dependencies, execution-order flexibility, and embodied feasibility.

To overcome these gaps, we introduce **RoboBench**, a benchmark systematically designed to evaluate MLLMs as the cognitive core for robotic manipulation. (1) **Comprehensive evaluation dimensions.** RoboBench defines five dimensions—Instruction Comprehension, Perception Reasoning, Generalized Planning, Affordance Prediction, and Failure Analysis—that together capture the interdependent capabilities required for embodied cognition. These dimensions come from the critical roles of the embodied brain in manipulation by tracing the full execution pipeline, shown in fig. 1. Before acting, it should interpret human intent in context Zhang et al. (2024). It then perceives the environment to gather task-relevant information. During execution, the brain decomposes long-horizon goals into sequential steps and adapts them in closed-loop. Each subgoal is further refined into spatially aligned cues to guide low-level controllers Yuheng Ji (2025); Team (2025). When failures occur, the system should diagnose, analyze and adjust errors to maintain robustness Duan et al. (2024). (2) **Realistic and diverse tasks.** We construct settings spanning single-arm, dual-arm, and mobile manipulation; objects with material attributes, physical properties, and world knowledge; multi-view

| Benchmarks | 🅰️ | ◑ | 🦥 | 🤖 | 👁 | ☝ | ❗ | ⌐L | 🏷 | 🗃 |
|---|---|---|---|---|---|---|---|---|---|---|
| RoboVQA Sermanet et al. (2024) | × | × | × | × | × | × | × | × | × | 21118 |
| EgoPlanBench Chen et al. (2024) | × | × | ✓ | × | × | ✓ | × | × | × | 4768 |
| MMRo Li et al. (2024d) | × | ✓ | ✓ | × | × | × | × | × | × | 26175 |
| EAI Li et al. (2024e) | × | × | ✓ | × | × | ✓ | × | × | ✓ | 438 |
| OpenEQA Majumdar et al. (2024) | × | ✓ | ✓ | × | × | × | × | × | × | 1636 |
| EgoThink Cheng et al. (2024b) | × | ✓ | ✓ | × | × | ✓ | × | × | × | 700 |
| VidEgoThink Cheng et al. (2024a) | × | ✓ | ✓ | × | × | ✓ | × | × | × | 4993 |
| EmbodiedEval Cheng et al. (2025) | × | ✓ | ✓ | × | × | × | × | × | ✓ | 328 |
| EmbodiedBench Yang et al. (2025) | ✓ | ✓ | ✓ | × | × | × | ✓ | × | ✓ | 1128 |
| VLABench Zhang et al. (2024) | ✓ | ✓ | ✓ | × | ✓ | ✓ | × | ✓ | ✓ | 2427 |
| RoboBench (**Ours**) | ✓ | ✓ | ✓ | ✓ | ✓ | ✓ | ✓ | ✓ | ✓ | 6077 |

Table 1: Comparison of RoboBench with existing benchmarks across key evaluation dimensions. From left to right, the icons in the header correspond to: 🅰️ - Language Instructions, ◑ - Commonsense Comprehension, 🦥 - Spatial Understanding, 🤖 - Embodiment Awareness, 👁 - Multi-view Consistency, ☝ - Affordance Reasoning, ❗ - Errors Analysis, ⌐L - Robustness Evaluation 🏷 - Annotations Broadness, 🗃 - Dataset Scale.

scenes with occlusion and memory-driven navigation Yang et al. (2024). Data combine large-scale real-world robotic datasets Fang et al. (2023); Collaboration et al. (2023); Liu et al. (2024c); Wu et al. (2024); Bu et al. (2025) with curated in-house collection, narrowing the sim-to-real gap. (3) **World-simulation rollout for planning evaluation.** Beyond symbolic matching, we evaluate long-horizon planning with a novel MLLM-as-world-simulator framework. Using the initial scene image, a ground-truth action list, and a human-annotated directed acyclic graph, evaluation MLLM builds structured task understanding, identifies critical object-state milestones, and rolls out predicted plans step by step under visual and physical constraints. This assesses both structural correctness and embodied feasibility—whether the plan could be executed successfully in the real world.

We evaluate 14 state-of-the-art MLLMs and uncover fundamental limitations: poor grounding of implicit instructions, fragile embodiment-specific and spatial-temporal perception, difficulty in long-horizon and cross-scenario planning, shallow fine-grained affordance understanding, and weak execution failure diagnosis. These findings show that while MLLMs hold promise as embodied brains, their reasoning remains superficial.

In summary, our contributions are as below:

- We introduce RoboBench, a comprehensive benchmark for evaluating MLLMs as embodied brains across 5 key dimensions, spanning 15 capabilities, 25 tasks, and 6077 QA pairs.

- We design realistic datasets and tasks across embodiments, objects, viewpoints, and task settings, providing precise reflections of real-world embodied interaction.

- We propose MLLM-as-world-simulator evaluation framework that faithfully assesses long-horizon planning by simulating whether plans achieve critical object-state milestones.

- We conduct large-scale systematic evaluation of SOTA MLLMs, establishing a leaderboard and offering actionable insights for advancing the embodied brain in robotic AI.

## 2 RELATED WORKS

Recent efforts have introduced several benchmarks for evaluating MLLMs in embodied AI tasks. Yet, significant limitations remain in assessing their role as the embodied brain as shown in Table 1. Many benchmarks Cheng et al. (2024a); Li et al. (2024b); Yang et al. (2024) lack semantically rich language instructions, limiting the ability to evaluate implicit task comprehension. Others Chi et al. (2024); Li et al. (2024d); Fan (2019) focus on a limited range of robotic embodiments, failing to assess generalization across single-arm, dual-arm, humanoid, and mobile robots. Additionally, most benchmarks overlook the evaluation of the multi-view planning Chen et al. (2024); Zheng et al. (2022); Fan (2019) and error analysis capabilities Cheng et al. (2024a; 2025); Majumdar et al. (2024), which are crucial for occlusion-aware planning and robust task execution, respectively. Finally, while some benchmarks Li et al. (2024e); Yang et al. (2025; 2024) include task planning, they often fail to account for variations in task execution order and textual granularity,

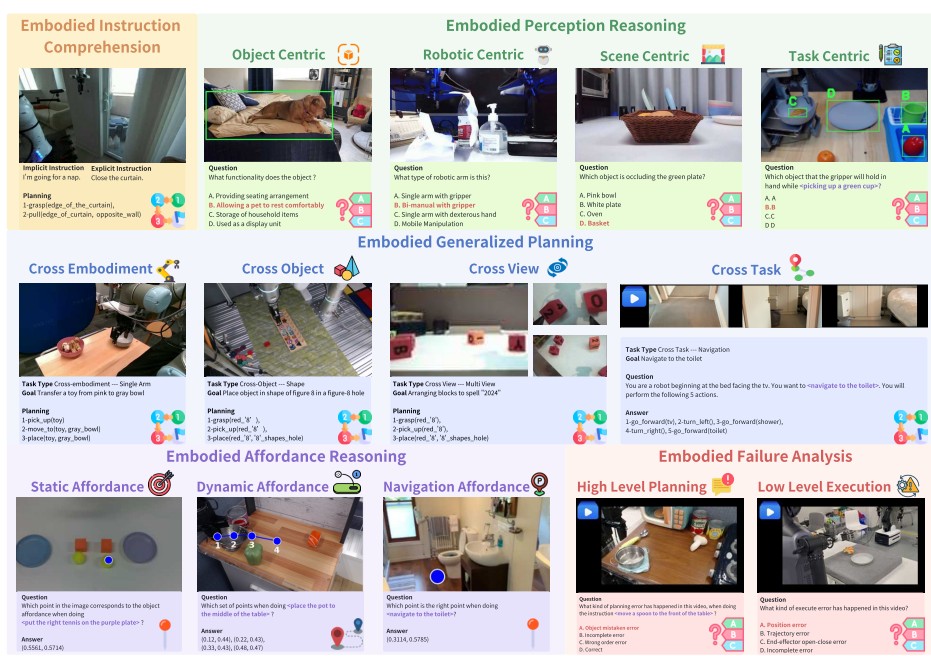

Figure 2: **Examples of RoboBench** Our benchmark covers 5 dimensions, 15 capabilities, and 25 tasks. We design diverse question formats, including multiple-choice, planning, and point prediction.

## 3 ROBOBENCH

### 3.1 CORE CAPABILITIES

Through a detailed analysis, we identify five critical dimensions of the MLLM-powered embodied brain, each aligned with the task execution pipeline: understanding human intent, perceiving the environment, formulating and adapting plans, refining actions via affordance prediction, and diagnosing failures. RoboBench evaluates these capabilities to uncover bottlenecks in embodied cognition.

**Embodied Instruction Comprehension**: *Can the embodied brain understand human intention?* Most embodied tasks rely on explicit instructions where actions and targets are clearly specified Sermanet et al. (2024); Yang et al. (2025); Li et al. (2025). In contrast, real-world instructions are often implicit (e.g., "I'm thirsty" instead of "Retrieve a drink") Zhang et al. (2024). This dimension evaluate whether models can infer both explicit and implicit instructions into actionable plans.

**Embodied Perception Reasoning**: *Can the embodied brain perceive the environment to gather task-relevant information?* Reliable planning and execution rely on accurate perception Majumdar et al. (2024); Li et al. (2024d), RoboBench evaluates this through four aspects: Robotic-centric considers embodiment type and viewpoint. Object-centric examines static and functional attributes. Scene-centric evaluate spatial relations, temporal grounding, and causality analysis. Task-centric assesses identifying instruction-relevant objects.

**Embodied Generalized Planning**: *Can the embodied brain generalize planning across embodiments, objects, views, and tasks?* Planning begins with decomposing long-horizon goals into subgoals, and continues during execution through predicting the next subtask, monitoring completion, and adapting subsequent steps Sermanet et al. (2024); Yang et al. (2025). We evaluates generalized planning across four aspects: embodiments (single-arm, dual-arm, mobile manipulator, and humanoid robot), objects (material attribute, physical attribute, and commonsense knowledge), views (multi-view integration under occlusion), and tasks (navigation planning using spatial cues from videos).

**Embodied Affordance Prediction**: *Can the embodied brain refine subtask plans through spatial affordances?* Beyond high-level planning, each subgoal should be transformed into spatial cues to guide low-level execution Liu et al. (2024a); Li et al. (2024c). Affordance prediction links subgoals

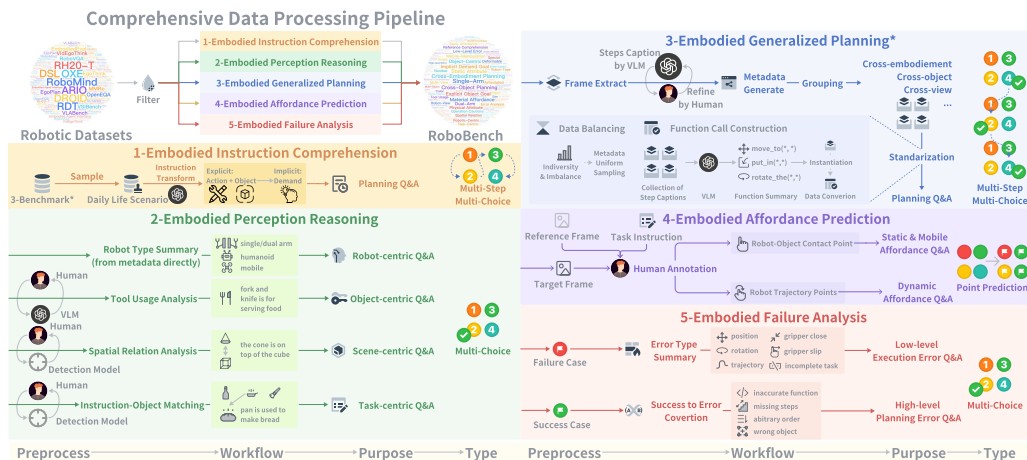

Figure 3: **Dataset Construction Pipeline.** RoboBench customizes data workflows for each dimension: orange for instruction, green for perception, blue for planning, purple for affordance, and red for reflection. Each workflow generally follows three stages—preprocessing, annotation with VLMs, detection models or human experts, and Q&A format generation.

with the embodiment, objects, and environment, enabling System 2 to instruct System 1 more effectively Li et al. (2024c). RoboBench evaluates three types: Static affordance identifys contact points (e.g., grasp an apple); Dynamic affordance predicts motion trajectories (e.g., open a drawer); Navigation affordance determines robot base position (e.g., approach a microwave on a distant table).

**Embodied Failure Analysis**: *Can the embodied brain detect and correct failures?* Open-world manipulation inevitably introduces errors, requiring the brain not only to identify them but also to diagnose causes and suggest corrections Liu et al. (2023b); Duan et al. (2024). RoboBench evaluates low-level execution errors (e.g., position misalignment, trajectory deviations, gripper failures, and incomplete actions), and high-level planning errors (e.g., wrong object, missing steps, incorrect ordering), providing insights for more robust and generalizable execution.

## 3.2 BENCHMARK CONSTRUCTION

**Dataset Collection and Processing Pipeline**    To evaluate the five cognitive dimensions, we build datasets from open-source and in-house robotic data, enriched with MLLM- and human-provided annotations, aligning each sub-benchmark with the embodied task pipeline for realism. The examples and construction pipeline are shown in fig. 2 and 3.

**Instruction Comprehension**: Evaluated via planning tasks, this dimension adopts a paired explicit–implicit setup. Explicit instructions are drawn from daily-life scenarios with clearly defined actions and targets, while implicit ones are generated by transforming them into demand-based requests using LLMs and human selection. This contrast tests the ability to interpret human intentions.

**Perception Reasoning**: Accurate perception is essential for reliable planning and execution. We construct datasets across four aspects: robotic-centric, using real robot data with type and view metadata for template-based QA; object-centric, combining curated static attributes Gao et al. (2024) with GPT-generated functional properties and distractors; scene-centric, leveraging Gemini-segmented video steps for temporal grounding and manual annotations of relative positions and key-point changes for spatial and causal reasoning; and task-centric, with human-labeled bounding boxes linking long-horizon instructions to target objects. All data are standardized into multiple-choice QA.

**Generalized Planning**: We build a planning pool from high-quality robotic videos Collaboration et al. (2023); Lin et al. (2024); Liu et al. (2024b;c); Fang et al. (2023); Wu et al. (2024); Mu et al. (2023); Yang et al. (2024), extracting frame sequences as standardized inputs. Gemini generates structured annotations—task summaries, step-wise instructions with timestamps, and metadata (objects, actions, scenes, embodiments)—which are refined by human annotators. Each step is then mapped into function templates (e.g., pick_up(object), move_to(object, target)) grouped into manipulation or

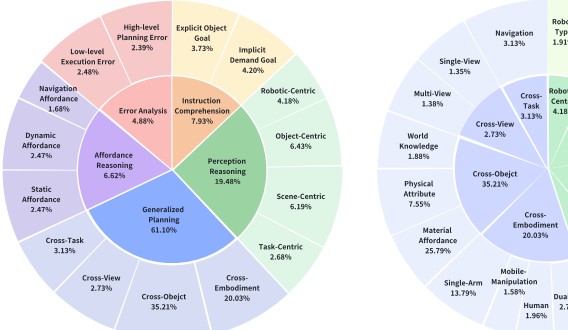

Figure 4: **Dimension Distribution of RoboBench.**

Table 2: **Dataset statistics.**

| Statistic | Number |
|---|---|
| Total items | 4333 |
| Total questions | 6077 |
| Perception and Failure questions | |
| Multiple-choice questions | 1480 |
| Instruction and Planning questions | |
| Q1 questions | 2171 |
| Q2 questions | 832 |
| Q3 questions | 1192 |
| Affordance questions | |
| Points questions | 252 |
| Points set questions | 150 |
| | |
| Instruction and Planning details-Q1 | |
| Avg. steps | 6.74 |
| Unique task instructions | 1466 |
| Unique answers | 1467 |
| | |
| Perception details | |
| Avg. question lengths | 131.3 |
| Avg. choice lengths | 13.8 |
| | |
| Failure details | |
| Avg. question lengths | 92.9 |
| Avg. choice lengths | 17.0 |

navigation Cheng et al. (2024a), guiding models toward structured plan generation. Evaluation covers three types Sermanet et al. (2024): (1) long-horizon planning, predicting the full action sequence from the first frame and instruction; (2) next-step planning, forecasting the (n+1)-th step from prior steps; and (3) task state estimation, deciding whether a given subtask has been completed.

**Affordance Prediction**: Affordances refine high-level subgoals into spatial cues for low-level execution Liu et al. (2024a). From the planning pool, representative frames are sampled and annotated with three types of affordance: static (contact points), dynamic (motion trajectories), and mobile (base positions). Models predict the corresponding point or trajectory given the task instruction.

**Failure Analysis**: It evaluates whether models can detect and reason about errors during execution Liu et al. (2023b); Duan et al. (2024). Execution-level failures (e.g., position misalignment, trajectory deviations, gripper errors) are collected from RoboMIND Wu et al. (2024) and labeled by experts, while planning-level failures are synthesized by perturbing correct instructions (wrong object, missing step, wrong order) due to lack of real-world planning failure data.

**Quality Control**   We adopt a two-stage quality control process to ensure benchmark quality: data filtering during construction and post-construction validation. In the construction stage, we apply both general and task-specific filters. General criteria address image quality and task validity. Sub-benchmarks also have tailored rules, such as excluding single-arm data from dual-arm tasks. After construction, professional researchers validate linguistic clarity, answerability, and correctness Liu et al. (2024d). We further adopt a majority-vote strategy Liu et al. (2024d): items all models solve correctly are removed, while items all models fail undergo manual review and correction.

**Dataset Statistics**   RoboBench consists of 6077 samples and 4333 unique items, providing a balanced mix of diversity and complexity to evaluate embodied brain capabilities. It covers 5 primary dimensions, 15 secondary meta-tasks, and 25 subtasks, ensuring comprehensive and challenging assessments. A detailed distribution was demonstrated in Figure 4. The detailed question statistics in different dimensions can be found in Table 2.

# 4 EVALUATION METRICS

To capture the diverse cognitive demands of embodied manipulation, we tailor metrics to each dimension. Perception reasoning and failure analysis are evaluated by multiple-choice accuracy Azzolini et al. (2025); Team et al. (2025). Affordance prediction adopts Euclidean distance for point prediction and RMSE for trajectory prediction Team (2025). Planning is assessed across three tasks (Q1–Q3). Q1 serves as the core, leveraging a visual–language world simulator to evaluate both structural fidelity and embodied feasibility, while Q2 and Q3 provide complementary perspectives.

**Q1 – Long-horizon planning.** Embodied planning decomposes long-horizon goals into atomic actions Cheng et al. (2024a). Each atomic action is defined by type and parameters (e.g., manipulated object, target), forming the nodes of our planning framework. To capture non-uniqueness in valid plans, we represent dependencies with a Directed Acyclic Graph (DAG) Zhang et al. (2024). Nodes

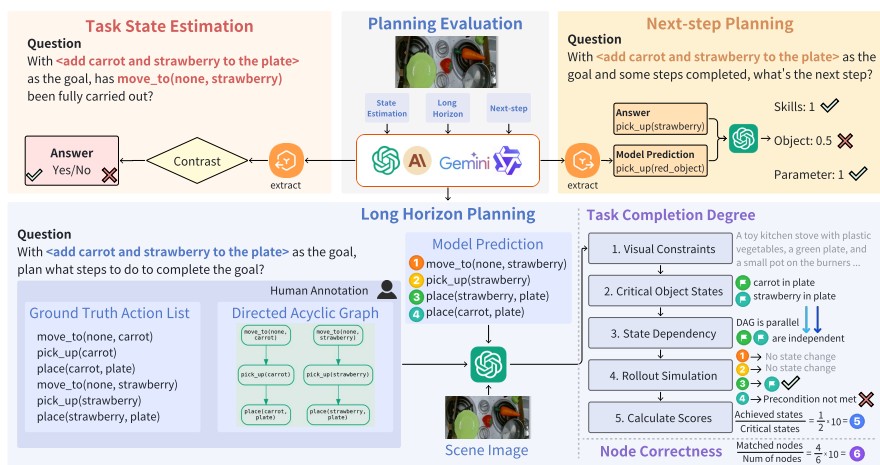

Figure 5: **Planning Evaluation Pipeline.** The planning benchmark includes three task types: long-horizon planning, next-step planning, and task state estimation. They are evaluated respectively with our proposed MLLM-as-world-simulator framework, direct LLM scoring, and binary accuracy.

correspond to atomic actions, and edges encode ordering constraints. The DAG highlights critical milestones (e.g., "drawer opened") while accommodating multiple valid execution orders. Manually annotated DAGs ensure quality and serve as reliable references for evaluation.

*Node correctness.* Each skill–object–parameter triple is treated as a node. The LLM judges a match by aligning skill, object, and parameter. The score is: NodeCorrectness $= \left\lfloor \frac{Y}{X} \times 10 \right\rfloor$, where $X$ is the number of ground-truth nodes and $Y$ the number of matches.

*Task completion via world-simulation rollout.* To move beyond symbolic matching, we simulate plan execution using three key inputs: (i) the first scene image for spatial and physical constraints, (ii) a ground-truth action list as a valid reference trajectory, and (iii) the annotated DAG encoding dependencies. The rollout proceeds as follows: (1) **Visual constraint analysis:** identify key objects, spatial relations, and physical constraints from the scene. (2) **Critical State change detection:** analyze object-level state transitions. (3) **Dependency enforcement:** use the DAG to check sequential/parallel relations between key states. (4) **Rollout simulation:** execute predicted actions step by step, checking preconditions, updating states, and recording achieved milestones. (5) **Scoring:** based on the proportion of critical states achieved: Completion $= \left\lfloor \frac{Y}{X} \times 10 \right\rfloor$, where $X$ is the number of critical states and $Y$ those successfully reached.

This simulator functions as a lightweight physics engine constrained by both language and vision, evaluating not just textual correctness but whether the plan could realistically succeed.

**Q2 – Next-step planning.** Given previous steps, the model predicts the next action. Scores are assigned for skill accuracy (0/1), object reasonableness (0/0.5/1), and parameter accuracy (0/0.5/1).

**Q3 – Task state estimation.** Q3 assesses whether the model correctly predicts subgoal completion, evaluated by binary accuracy.

## 5 EXPERIMENT

We evaluate several MLLMs on RoboBench, including closed-source MLLMs, open-source multi-image MLLMs, and open-source embodied MLLMs. Detailed descriptions of the models are provided in the supplementary materials. Additionally, we introduce text-only LLM and human as reference.

### 5.1 OVERALL RESULTS

**Gemini-2.5-Pro leads but still lags behind humans**: Among all evaluated models, Gemini-2.5-Pro shows the most consistent advantages across dimensions, clearly outperforming both other closed-source and open-source MLLMs. However, a notable gap to human-level performance remains,

Table 3: **Results on Perception Reasoning(%)**Attr. stand for Attribute; Temp. stand for Temporal;

| Model | Robotic-centric | | Object-centric | | Scene-centric | | | Task-centric | Avg |
|---|---|---|---|---|---|---|---|---|---|
| | Robot-type | Robot-view | Static Attr. | Functional Attr. | Spatial Relation | Temp. Grounding | Causality | Refer. Comprehen. | |
| **Basic Reference** | | | | | | | | | |
| Human Evaluation | 80.67 | 79.08 | 43.77 | 83.89 | 70.91 | 51.61 | 91.22 | 93.22 | 74.30 |
| GPT-4o-text-only | 20.51 | 13.77 | 5.18 | 35.37 | 25.74 | 18.32 | 25.52 | 22.09 | 20.81 |
| **Closed-Source MLLMs** | | | | | | | | | |
| GPT-4o-Mini | 38.75 | 18.84 | 26.43 | 53.66 | 30.36 | 22.65 | 34.25 | 39.67 | 33.08 |
| GPT-4o | **64.96** | 39.38 | 24.92 | 46.75 | 42.24 | 20.61 | 33.10 | 41.31 | 39.16 |
| Claude-3.5-Sonnet | 41.31 | 36.23 | 29.13 | 62.60 | 34.98 | 21.88 | 36.09 | 25.36 | 35.95 |
| Claude-3.7-Sonnet | 40.46 | 32.37 | 45.20 | 71.14 | 36.63 | 21.09 | 40.92 | 28.02 | 39.48 |
| Gemini-2.0-Flash | 56.69 | 20.77 | 49.08 | 78.46 | 42.57 | 21.37 | 51.72 | 72.40 | 49.13 |
| Gemini-2.5-Flash | 62.39 | 39.38 | **55.02** | 77.24 | 57.43 | 33.58 | 70.34 | 74.64 | 58.75 |
| Gemini-2.5-Pro | 64.30 | 41.71 | 54.83 | **82.27** | 60.44 | 49.68 | 71.73 | 78.68 | 62.96 |
| Qwen-VL-Plus | 28.21 | 21.74 | 34.63 | 58.54 | 27.72 | 21.37 | 31.03 | 34.36 | 32.20 |
| Qwen-VL-Max | 47.86 | **43.48** | 39.70 | 75.20 | 50.17 | 27.45 | 37.93 | 41.53 | 45.42 |
| **Open-Source Multi-Image MLLMs** | | | | | | | | | |
| LLaVA-OneVision-0.5B | 30.34 | 23.68 | 37.08 | 49.66 | 27.27 | 18.42 | 23.65 | 19.21 | 28.66 |
| LLaVA-OneVision-7B | 44.83 | 30.26 | 33.43 | 75.84 | 45.45 | 23.68 | 25.68 | 44.63 | 40.48 |
| Qwen2.5-VL-7B-Ins | 23.93 | 26.81 | 37.86 | 46.34 | 31.68 | 22.90 | 14.48 | 36.81 | 30.10 |
| Qwen2.5-VL-72B-Ins | 47.72 | 42.75 | 41.74 | 72.95 | 48.51 | 27.87 | 40.32 | 42.13 | 45.50 |
| **Embodied MLLMs** | | | | | | | | | |
| RoboBrain-2.0-7B | 44.97 | 24.84 | 40.43 | 79.19 | 48.18 | 23.48 | 41.22 | 53.67 | 44.50 |

Table 4: Results on Instruction Comprehension and Generalized Planning

| Model | Instruction Comprehension | | | Generalized Planning | | | | | | | | | | |
|---|---|---|---|---|---|---|---|---|---|---|---|---|---|---|
| | | | | Cross-Embodiment Planning | | | | Cross-Object Planning | | | Cross-View Planning | | Cross-Task Planning | Avg |
| | Explicit | Implicit | Avg | Single-arm | Dual-arm | Mobile-manip. | Human | Material Afford. | Physical Attr. | World Knowl. | Multi | Single | Navigation Plan. | |
| **Basic Reference** | | | | | | | | | | | | | | |
| Human Evaluation | 59.94 | 61.13 | 60.54 | 72.50 | 41.93 | 41.55 | 62.28 | 56.70 | 58.98 | 49.36 | 52.82 | 51.59 | 45.23 | 54.50 |
| GPT-4o-text-only | 38.80 | 11.10 | 24.95 | 26.70 | 33.32 | 43.65 | 37.86 | 36.58 | 22.33 | 37.68 | 44.35 | 38.11 | 36.90 | 33.95 |
| **Closed-Source MLLMs** | | | | | | | | | | | | | | |
| GPT-4o-Mini | 41.21 | 14.95 | 28.08 | 27.47 | 25.21 | 37.98 | 31.72 | 38.46 | 42.56 | | 39.11 | 33.29 | 34.04 | 33.31 |
| GPT-4o | 45.60 | 19.04 | 32.32 | 28.28 | 32.65 | **52.69** | 35.71 | 39.93 | 46.09 | 41.34 | 38.51 | 33.66 | 39.41 | 37.74 |
| Claude-3.5-Sonnet | 42.11 | 14.85 | 28.48 | **30.18** | 33.65 | 50.29 | **41.05** | 38.28 | 40.67 | 39.63 | 45.95 | 40.43 | 39.77 | 38.07 |
| Claude-3.7-Sonnet | 47.77 | 14.53 | 31.15 | 29.86 | 38.69 | 50.39 | 37.06 | 38.65 | 41.86 | 51.83 | 48.19 | 44.51 | 39.95 | 41.68 |
| Gemini-2.0-Flash | 43.49 | 16.38 | 29.93 | 28.67 | 33.66 | 48.27 | 33.95 | 40.76 | 54.27 | 40.12 | 46.13 | 40.73 | 37.02 | 38.62 |
| Gemini-2.5-Flash | 42.53 | 17.10 | 29.82 | 27.05 | 40.46 | 49.91 | 34.50 | 39.87 | 53.37 | 46.22 | 39.41 | 43.29 | 38.32 | 39.33 |
| Gemini-2.5-Pro | 51.15 | 19.60 | 35.37 | 29.71 | 37.65 | 50.96 | 37.44 | 39.29 | 56.50 | 43.29 | 47.35 | 45.12 | 43.62 | 41.81 |
| Qwen-VL-Plus | 37.77 | 10.38 | 24.07 | 24.68 | 21.75 | 32.98 | 33.91 | 28.45 | 33.55 | 33.78 | 30.95 | 28.60 | 4.39 | 26.77 |
| Qwen-VL-Max | 46.45 | 16.98 | 31.71 | 28.30 | 35.73 | 47.79 | 32.40 | 40.44 | 44.33 | 42.32 | 41.79 | 37.68 | 38.00 | 37.68 |
| **Open-Source Multi-Image MLLMs** | | | | | | | | | | | | | | |
| LLaVA-OneVision-0.5B | 6.82 | 1.24 | 3.61 | 2.90 | 4.57 | 4.77 | 3.68 | 4.77 | 3.47 | 6.47 | 4.30 | 3.62 | 11.39 | 4.83 |
| LLaVA-OneVision-7B | 18.93 | 3.48 | 10.05 | 11.48 | 16.23 | 8.27 | 5.34 | 18.51 | 15.62 | 8.10 | 0.00 | 15.16 | 24.67 | 12.15 |
| Qwen2.5-VL-7B-Ins | 26.45 | 4.65 | 15.55 | 19.47 | 12.90 | 28.75 | 28.19 | 22.06 | 21.63 | 25.61 | 11.79 | 20.12 | 2.10 | 18.64 |
| Qwen2.5-VL-72B-Ins | 46.81 | 15.15 | 30.98 | 28.20 | 36.92 | 49.14 | 31.31 | 40.51 | 44.94 | 38.90 | 43.16 | 40.24 | 37.47 | 37.73 |
| **Embodied MLLMs** | | | | | | | | | | | | | | |
| RoboBrain-2.0-7B | 36.93 | 8.19 | 22.51 | 15.46 | 25.32 | 32.72 | 31.81 | 19.85 | 30.85 | 23.24 | 31.51 | 23.89 | 24.53 | 25.35 |

underscoring the difficulty of achieving robust embodied intelligence. Other MLLMs demonstrate highly uneven or generally weak capabilities in embodied tasks.

**Closed-source models outperform open-source models**: Closed-source models maintain a clear lead in four of the five dimensions, with performance margins often reaching 10–15%. Only in perception reasoning do open-source models approach comparable results, while gaps in instruction comprehension, planning, affordance reasoning, and failure analysis remain substantial.

**Performance varies across dimensions**: RoboBench results reveal large differences across task categories. Perception reasoning achieves relatively higher accuracy, while generalized planning remains the most challenging. These disparities reflect the varying cognitive demands of each dimension and the limits of current evaluation, highlighting where future progress is most needed.

## 5.2 Fine-grained Results across Sub-Tasks

We further break down results across sub-tasks, highlighting distinct weaknesses in instruction comprehension, perception, planning, and failure analysis.

**Implicit vs. Explicit Instruction Understanding**: Models perform significantly worse on implicit instructions compared to explicit ones, even when images and ground truth remain the same. This indicates a clear weakness in grounding indirect human demands into actionable goals. A likely reason is that current MLLMs fail to fully integrate scene context when inferring task intent. Future improvements require models to jointly consider language, perception, and context in order to accurately infer and execute human intentions.

Table 5: Results on Planning Q2, Q3.

| Model | Instr. Compre. | | Generalized Planning | | | | | |
| --- | --- | --- | --- | --- | --- | --- | --- | --- |
| | Explicit Goal | | Single Arm | | Mater. Afford. | | World Knowl. | |
| | Q2 | Q3 | Q2 | Q3 | Q2 | Q3 | Q2 | Q3 |
| Basic Reference | | | | | | | | |
| Human Evaluation | 45.28 | 74.32 | 27.52 | 71.35 | 43.62 | 71.2 | 43.89 | 69.83 |
| GPT-4o-text-only | 33.07 | 56.25 | 17.22 | 51.77 | 29.27 | **51.83** | 33.33 | 51.22 |
| Closed-Source MLLMs | | | | | | | | |
| GPT-4o-Mini | 39.06 | 61.25 | 22.35 | 52.32 | 31.56 | 49.13 | **44.27** | 36.59 |
| GPT-4o | 41.41 | 62.50 | 21.64 | 57.49 | 31.92 | 51.87 | 42.19 | 46.34 |
| Claude-3.5-Sonnet | 33.86 | 55.00 | 21.56 | 57.49 | 29.60 | 52.54 | 39.60 | 46.34 |
| Claude-3.7-Sonnet | 38.54 | 61.25 | 22.88 | 59.13 | 30.98 | 57.88 | 32.29 | 60.98 |
| Gemini-2.0-Flash | 41.93 | 68.75 | **26.27** | 63.49 | 34.98 | 61.95 | 42.19 | 56.10 |
| Gemini-2.5-Flash | 40.89 | 71.25 | 24.28 | **72.21** | 33.46 | 69.39 | 41.15 | 60.98 |
| Gemini-2.5-Pro | 40.10 | **75.00** | 25.09 | 70.30 | **36.48** | **69.67** | 33.85 | **65.85** |
| Qwen-VL-Plus | 41.15 | 52.50 | 20.65 | 54.22 | 29.27 | 50.49 | 39.58 | 51.22 |
| Qwen-VL-Max | **44.27** | 66.25 | 23.31 | 65.94 | 31.72 | 62.76 | 35.94 | 53.66 |
| Open-Source Multi-Image MLLMs | | | | | | | | |
| LLaVA-OneVision-0.5B | 9.42 | 42.50 | 15.91 | 45.23 | 10.77 | 49.84 | 5.92 | 41.46 |
| LLaVA-OneVision-7B | 40.26 | 58.75 | 53.10 | 52.04 | 44.54 | 51.95 | 43.42 | 60.98 |
| Qwen2.5-VL-7B-Ins | 22.92 | 55.00 | 12.87 | 53.13 | 17.30 | 49.41 | 18.75 | 51.22 |
| Qwen2.5-VL-72B-Ins | 42.45 | 61.25 | 26.19 | 60.76 | 33.56 | 52.08 | 32.81 | 48.78 |
| Embodied MLLMs | | | | | | | | |
| RoboBrain-2.0-7B | 36.17 | 53.75 | 15.36 | 54.28 | 23.62 | 49.58 | 28.53 | 50.26 |

Table 6: Results on Afford. Pred. and Failure.

| Model | Affordance Prediction | | | | Failure Analysis | | |
| --- | --- | --- | --- | --- | --- | --- | --- |
| | Static | Dynamic | Naviga. | Avg | Execution | Planning | Avg |
| Basic Reference | | | | | | | |
| Human Evaluation | 86.08 | 80.02 | 81.85 | 82.63 | 47.30 | 80.67 | 63.99 |
| GPT-4o-text-only | 44.89 | 40.70 | 38.19 | 39.88 | 25.17 | 37.93 | 31.55 |
| Closed-Source MLLMs | | | | | | | |
| GPT-4o-Mini | 50.64 | 42.88 | 42.30 | 46.39 | 17.66 | 44.60 | 31.13 |
| GPT-4o | 55.61 | 49.14 | 49.91 | 51.91 | 22.29 | 57.01 | 39.65 |
| Claude-3.5-Sonnet | 56.26 | 54.25 | 53.84 | 54.77 | 16.12 | 47.52 | 31.82 |
| Claude-3.7-Sonnet | 60.02 | 52.38 | 50.07 | 54.06 | 18.32 | 54.24 | 36.28 |
| Gemini-2.0-Flash | 61.65 | 61.76 | 66.89 | 63.37 | **28.48** | 59.80 | 44.14 |
| Gemini-2.5-Flash | 61.20 | 52.04 | 52.01 | 54.29 | 18.54 | 67.65 | 43.1 |
| Gemini-2.5-Pro | 70.54 | 62.03 | 63.96 | 65.21 | 15.96 | **74.31** | **45.14** |
| Qwen-VL-Plus | 51.74 | 37.42 | 47.97 | 48.18 | 13.91 | 40.00 | 26.96 |
| Qwen-VL-Max | 70.01 | 56.26 | 50.85 | 59.43 | 17.22 | 57.93 | 37.58 |
| Open-Source Multi-Image MLLMs | | | | | | | |
| LLaVA-OneVision-0.5B | 20.56 | 28.56 | 27.69 | 24.76 | 21.19 | 24.67 | 22.93 |
| LLaVA-OneVision-7B | 23.83 | 33.61 | 33.43 | 30.29 | **29.14** | 34.00 | 31.56 |
| Qwen2.5-VL-7B-Ins | 49.73 | 38.03 | 42.16 | 43.15 | 13.91 | 26.90 | 20.41 |
| Qwen2.5-VL-72B-Ins | **71.54** | 51.94 | 47.67 | 56.67 | 12.59 | 50.72 | 31.66 |
| Embodied MLLMs | | | | | | | |
| RoboBrain-2.0-7B | 51.87 | 54.63 | 41.61 | 49.37 | 7.95 | 42.00 | 41.24 |

**Perception Challenges**: While models show reasonable performance in object property analysis, they struggle with basic robotic perception and spatiotemporal reasoning. Common failures include misidentifying robot type and viewpoint, or failing to reason under different reference frames. Moreover, causal reasoning across time remains weak. These gaps suggest that future models should incorporate stronger embodiment-aware perception modules and explicit spatial–temporal reasoning.

**Planning Limitations**: Models exhibit limited capacity in complex planning scenarios. Cross-Embodiment: Current models, largely trained on single-arm settings, often fail to coordinate dual-arm actions or perform mobile manipulation, leading to poor spatial search and movement decisions. Cross-Object: Performance drops sharply when tasks involve uncommon objects, symbolic reasoning, or world knowledge, suggesting weak integration of heterogeneous information, even though results on common objects remain acceptable. Cross-View: Under occlusion, models fail to exploit multi-view inputs effectively, showing little gain, which highlights the need for better multi-view reasoning.

**Error Analysis**: Analyzing execution errors proves harder than diagnosing planning errors, with scores consistently lower on the former. Execution-level failures often require fine-grained distinctions—such as separating position errors from rotation errors (correct position but wrong gripper angle)—which demand expert-level embodied knowledge. Human performance is also relatively poor in such cases, underscoring the intrinsic difficulty. This further suggests that enhancing fine-grained perception is crucial for improving models' error diagnosis capabilities.

## 5.3 PLANNING ERROR ANALYSIS

Our evaluations reveal several common errors in MLLMs. **Missing Steps**: Skipping essential actions, leading to incomplete task execution. **Wrong Objects**: Output involves an object unrelated to the task. **Physics Violations**: Physically impossible steps are included. **Spatial Reasoning Errors**: Failure in spatial or directional reasoning. These errors highlight the limitations of MLLMs.

## 6 CONCLUSION

We presented RoboBench, a benchmark to systematically evaluate MLLMs as the embodied brain for robotic manipulation. It traces the full execution pipeline across five dimensions—Instruction Comprehension, Perception Reasoning, Generalized Planning, Affordance Prediction, and Failure Analysis—covering 25 tasks with over 6K QA pairs. A core contribution is a world-simulator framework that evaluates not only symbolic plan fidelity but also embodied feasibility. Experiments expose major gaps in implicit instruction grounding, perception, long-horizon planning, affordance reasoning, and failure diagnosis. RoboBench thus provides a unified scaffold for measuring embodied cognition and guiding the development of more robust, generalizable embodied intelligence.

## 7 ETHICS STATEMENT

This study investigates multimodal large language models (MLLMs) as the embodied brain for robotic manipulation. All experiments are conducted in controlled simulation environments and robotic platforms, without involving human subjects, crowdsourced annotators, or personally identifiable information. The datasets used are either publicly available (e.g., large-scale robotic demonstrations, human motion capture databases) or collected in-house under authorized settings. Our benchmark is intended to advance embodied intelligence for socially beneficial applications such as assistive robotics and human–robot collaboration. At the same time, we recognize potential risks if unaligned embodied agents are deployed in real-world environments. To mitigate these risks, RoboBench restricts evaluation to cognitive capabilities and employs world-simulator rollout under physical and visual constraints, avoiding uncontrolled deployment. We are committed to developing safe, fair, and transparent embodied AI, and declare no competing interests.

## 8 REPRODUCIBILITY STATEMENT

Reproducibility is a central design principle of RoboBench. We provide complete implementation details—including dataset demos, automatic annotation prompt for different questions, and evaluation protocols—in the main text and appendix. All motion sequences and task datasets are either publicly available or released alongside this work. We further establish a standardized evaluation pipeline and leaderboard to facilitate consistent comparison across models. Upon acceptance, we will release the benchmark datasets, evaluation scripts, and codebase to enable faithful replication and extension by the community.

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

## A  APPENDIX OVERVIEW

This appendix provides additional materials that complement the main paper, covering the use of LLMs, examples, baselines, extended analyses, statistics, sensitivity studies, prompts, and case studies:

- **Use of LLMs (Appendix B)**: Description of how large language models were leveraged throughout coding, debugging, and manuscript polishing stages to assist with technical guidance and language refinement.

- **Detailed Examples (Appendix C)**: Representative instances are provided together with a compressed archive to facilitate reproducibility.

- **Baseline Models (Appendix D)**: Configurations and comparative results of large-scale foundation models under adjusted evaluation settings.

- **Extended Experiment Analysis (Appendix E)**: Comprehensive tables and in-depth error analyses that complement the main findings.

- **Statistics (Appendix F)**: Dataset source breakdowns, affordance prediction visualizations, and planning complexity distributions.

- **Case Study (Appendix G)**: Qualitative examples contrasting correct and incorrect model reasoning, with error type annotations and visual illustrations.

- **Prompts (Appendix H)**: Representative prompts used during data generation, model inference, and evaluation, enabling transparency and reproducibility.

## B  THE USE OF LLMS

During the development of this work, we employed large language models (LLMs) in two complementary ways. First, LLMs were consulted during the coding and debugging process to provide technical suggestions and accelerate implementation. Second, following the collaborative preparation of the manuscript by the authors, LLMs were used to refine the clarity, fluency, and overall presentation of the text. All conceptual contributions, experimental designs, and substantive analyses were solely the responsibility of the authors.

## C  DETAILED EXAMPLES

To promote transparency and reproducibility, we provide a curated set of fine-grained examples from ROBOBENCH, covering all sub-benchmarks, meta-tasks, and sub-tasks. Each example illustrates the task formulation with detailed explanations, problem categories, and representative input–output pairs. The complete collection is available in the supplementary file `robobench_demo.zip`.

## D  BASELINE MODELS

To ensure fair and comprehensive evaluation, we benchmarked against a diverse set of baseline models spanning both open- and closed-source systems. Table 7 provides a systematic overview of these models, including their accessibility (open vs. closed source), creators, and corresponding API endpoints or model identifiers.

## E  MORE EXPERIMENT ANALYSIS

### E.1  ERROR DEFINITIONS AND ADDITIONAL ANALYSIS

#### E.1.1  ERROR TAXONOMY

In this section, we define the types of errors and sub-errors encountered in our benchmark. We categorize errors into four main types: *Execution Errors*, *Identification Errors*, *Common Sense Errors*, and *Mode-Specific Errors*. Each error type corresponds to a distinct failure mode of the

Table 7: The specific information of models in Experimental Results

| Model Name | Type | Creator | API Name/Huggingface Model ID |
|---|---|---|---|
| GPT-4o | Closed-Source | OpenAI | gpt-4o-2024-11-20 |
| GPT-4o-mini | Closed-Source | OpenAI | gpt-4o-mini-2024-07-18 |
| Gemini-2.5-Pro | Closed-Source | Google | gemini-2.5-pro |
| Gemini-2.5-Flash | Closed-Source | Google | gemini-2.5-flash |
| Claude-3.7-Sonnet | Closed-Source | Anthropic | claude-3.7-sonnet-20250219 |
| Claude-3.5-Sonnet | Closed-Source | Anthropic | claude-3.5-sonnet-20241022 |
| Qwen-VL-Max | Closed-Source | Qwen | Qwen-VL-Max |
| Qwen-VL-Plus | Closed-Source | Qwen | Qwen-VL-Plus |
| Qwen2.5-VL-72B-Ins | Open-Source | Qwen | Qwen/Qwen2.5-VL-72B-Instruct |
| Qwen2.5-VL-7B-Ins | Open-Source | Qwen | Qwen/Qwen2.5-VL-7B-Instruct |
| LLaVA-OneVision-0.5B | Open-Source | NTU | lmms-lab/llava-onevision-qwen2-0.5b-ov |
| LLaVA-OneVision-7B | Open-Source | NTU | lmms-lab/llava-onevision-qwen2-7b-ov |
| RoboBrain-7B-2.0 | Open-Source | BAAI | BAAI/RoboBrain2.0-7B |

embodied brain. For instance, execution errors arise when generating or executing action sequences, identification errors occur when associating objects or parameters, common sense errors reflect violations of physical or spatial priors, and mode-specific errors are tied to task-specific formatting requirements. A detailed taxonomy of error types and definitions is provided in Table 8.

| Error Type | Sub-error | Definition |
|---|---|---|
| Execution Errors | Missing Steps | The predicted action list omits necessary functions for task completion. |
| | Impossible Actions | The model outputs action functions that are not defined in the action space. |
| | Redundant Steps | The model outputs an excessive number of action functions beyond requirements. |
| | Wrong Function | The model selects an incorrect action function for the intended operation. |
| Identification Errors | Aliasing Errors | Failing to distinguish between visually similar objects (e.g., recognizing a crumpled paper ball as popcorn). |
| | Parameter Mismatch | Producing incorrect parameter values for otherwise valid functions. |
| | Wrong Object | Referring to objects completely unrelated to the intended target. |
| Common Sense Errors | Physics Violations | Generating physically impossible action sequences (e.g., folding multiple clothes simultaneously). |
| | Spatial Reasoning Errors | Failing to infer correct spatial operations (e.g., twisting a faucet in the wrong direction). |
| Mode-Specific Errors | CSS Mode Reference Errors | Referring to objects directly with natural language descriptions instead of the required object indices. |
| | Dual Arm Assignment | In dual arm tasks, the robot fails to use the arm that is closer to the target object, resulting in an unreasonable operation. |

Table 8: Error taxonomy with definitions in RoboBench.

### E.1.2 Error Analysis

**Execution Errors.** This category reflects failures in generating valid action sequences. We identify four common subtypes: (1) *Missing Steps*: the predicted sequence omits essential actions, such as failing to include a "pick up object" step, which prevents successful task completion. (2) *Impossible Actions*: the model produces functions not defined in the available action space, e.g., outputting an undefined "teleport" action. (3) *Redundant Steps*: the model generates more actions than required,

often inserting unnecessary "move" operations that deviate from the ground truth sequence. (4) *Wrong Function*: the model selects the incorrect action for the intended goal, such as using "push" instead of "pull."

**Identification Errors.** These errors occur when the model fails to correctly associate actions with the appropriate objects or parameters. (1) *Aliasing Errors*: the model confuses visually similar objects, such as misidentifying a crumpled paper ball as popcorn. (2) *Parameter Mismatch*: although the correct function is chosen, its parameters are incorrect, e.g., attempting to grasp an object with an inappropriate force setting. (3) *Wrong Object*: the output involves manipulating an irrelevant object, such as operating on a cup when the instruction specifies a book.

**Common Sense Errors.** These errors reveal violations of physical feasibility or basic spatial reasoning. (1) *Physics Violations*: the predicted plan contains physically impossible steps, such as folding multiple pieces of clothing simultaneously. (2) *Spatial Reasoning Errors*: the model fails to infer correct spatial or directional relations, e.g., twisting a faucet in the wrong direction.

**Mode-Specific Errors.** Certain failures are tied to formatting or mode-specific requirements. (1) *CSS Mode Reference Errors*: instead of adhering to symbolic references, the model outputs natural language descriptions of objects, such as predicting "the red cup" rather than the required identifier "Object_3."

Overall, these errors highlight distinct limitations across execution, perception, reasoning, and adherence to task-specific conventions. Addressing them requires better grounding of action functions, stronger disambiguation of objects, improved spatial reasoning, and stricter alignment with mode constraints.

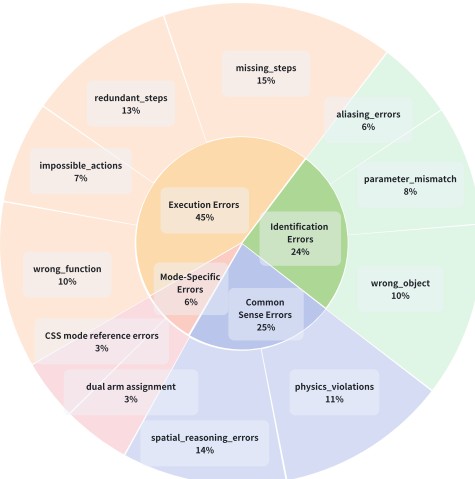

Figure 6: **Error analysis.** Proportional distribution of different error types.

# F MORE STATISTICS OF PLANNING TASKS

This section provides additional statistics for planning tasks in **RoboBench**, covering detailed distributions of skills, actions, objects, instructions, and task dimensions, as well as dataset source distributions.

## F.1 DETAILED TASK STATISTICS

**Skill Statistics** Figure 8 presents the distribution of skills. The horizontal axis shows skill names, and the vertical axis indicates counts.

**Action Statistics** Figure 9 shows the distribution of action sequence lengths across tasks, while Figure 10 shows action name vs. frequency counts.

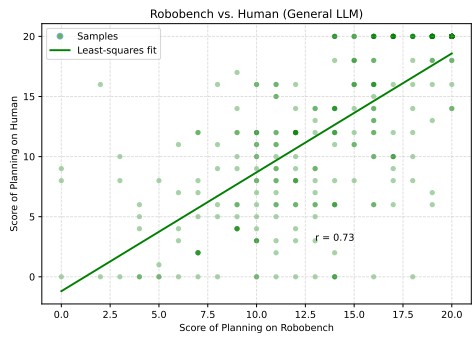 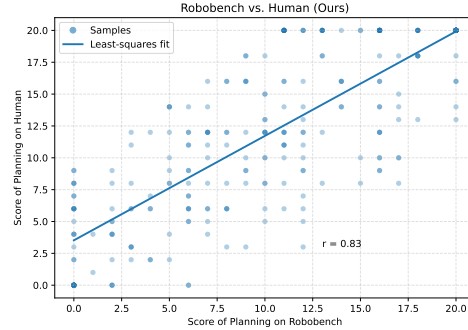

(a) Baseline evaluation framework.

(b) MLLM-as-world-simulator evaluation.

Figure 7: Comparison of how closely our proposed evaluation method and the baseline align with human evaluation results. Pearson correlation coefficient r closer to 1 indicates stronger alignment with human evaluation results.

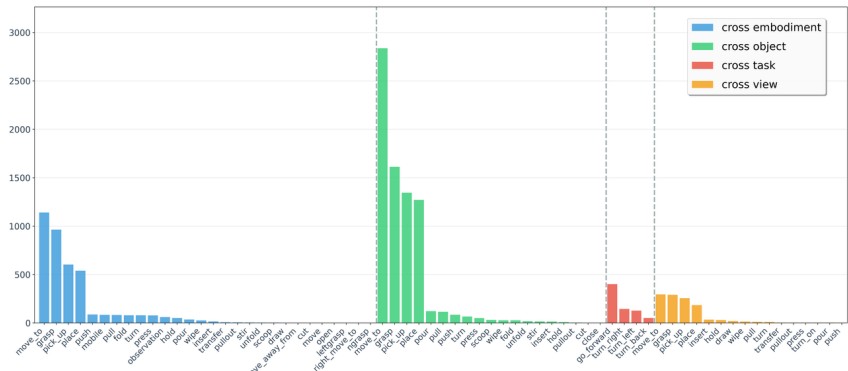

Figure 8: Skill distributions in RoboBench planning tasks. unique skill counts per data entry.

## F.2 DATASET SOURCE DISTRIBUTION

Figure 11 presents the distribution of source datasets for planning tasks in RoboBench. The figure shows the proportion of tasks contributed by each source.

## G CASE STUDY

To provide further insights, we present illustrative case studies of our evaluation. Figure 12 shows representative error cases where the model fails to reason correctly, while Figure 13 highlights successful examples that demonstrate the model's robustness.

## H PROMPTS

This section presents the prompts employed for both **benchmark data construction** and **model evaluation** in RoboBench. We organize them by task type and processing stage, emphasizing reproducibility and precision. All prompt content placeholders are left empty for subsequent insertion.

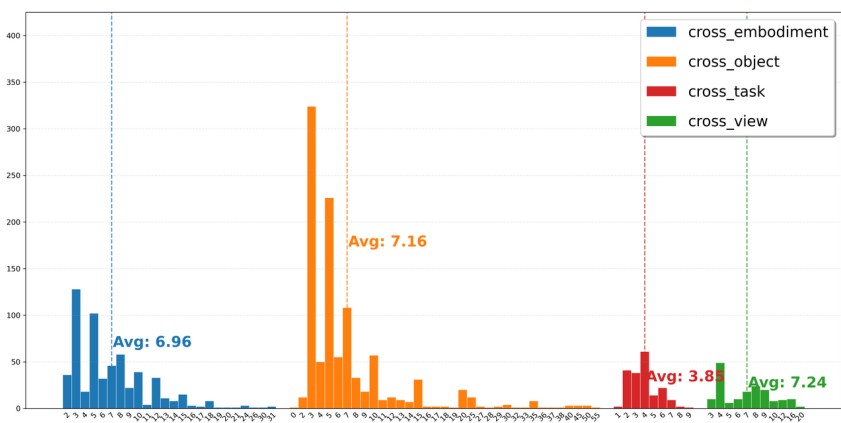

Figure 9: Distribution of action sequence lengths in RoboBench planning tasks.

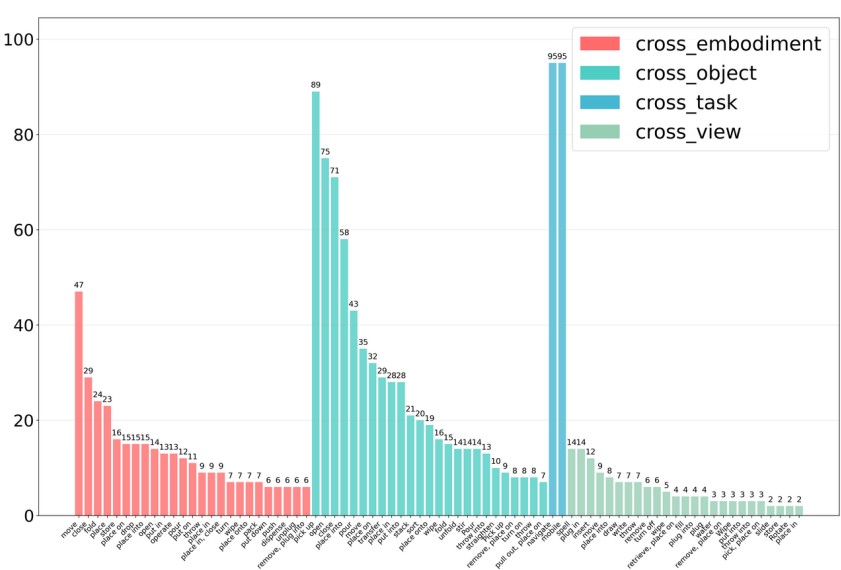

Figure 10: Frequency of each action name in RoboBench planning tasks.

## H.1 BENCHMARK DATA CONSTRUCTION

### H.1.1 PLANNING TASKS

**1. Video to Natural Language Task Description**    Robot videos are converted into natural language task descriptions, including sequential step descriptions. Three prompts are used to cover different robot types: single-arm, dual-arm, and mobile-manipulator robots, shown in Figure 14, 15 and 16.

**2. NL Steps → Predefined Function Sequence**    Natural language step descriptions are mapped to a sequence of predefined functions. Two function lists are used: Manipulation (Figure 17) and Navigation (Figure 18), followed by a conversion prompt (Figure 19) referencing these lists.

**3. Function Instantiation**    Instantiate function arguments with objects extracted from step descriptions. The prompt is presented in Figure 20.

**4. Explicit → Implicit Instruction Conversion**    To facilitate natural language task formulation for embodied robots, we convert explicit task instructions into implicit forms that imply the required action without directly naming the target object or task (Figure 21).

Figure 11: Source distribution of planning tasks in RoboBench dataset.

**5. Fine-grained Attribute Extraction**   We design prompts to extract fine-grained, step-level information from video frames, including **objects** (Figure 22), **actions** (Figure 23), and **scene labels** (Figure 24). These structured annotations provide the foundation for evaluating compositional reasoning and downstream task performance.

### H.1.2   PERCEPTION TASKS: FUNCTIONAL ATTRIBUTE SUBTASKS

To evaluate an AI system's understanding of object functionality in images, we generate multi-choice question-answer pairs based on objects highlighted with bounding boxes, focusing on detailed attribute analysis and functional reasoning (Figure 25).

### H.2   EVALUATION PROMPTS

### H.2.1   PLANNING TASKS: Q1 EVALUATION

For Q1-type planning tasks, we provide a structured two-step evaluation: first, extracting the action sequence from model responses (Figure 26), and second, scoring the extracted plans against ground truth data using embodied task reasoning criteria (Figure 27).

### H.2.2   PLANNING TASKS: Q2 EVALUATION

For Q2-type planning tasks, we follow a two-stage procedure: first, extracting the next actionable step from the model response in a structured robotic action format (Figure 28), and second, quantitatively scoring the extracted step against the ground truth using strict skill, object, and parameter evaluation criteria (Figure 29).

### H.2.3   PLANNING TASKS: Q3 EVALUATION

For Q3-type planning tasks, the evaluation involves converting model responses into a strict binary decision, where only "yes" or "no" is returned without any additional text or formatting, as illustrated in Figure 30.

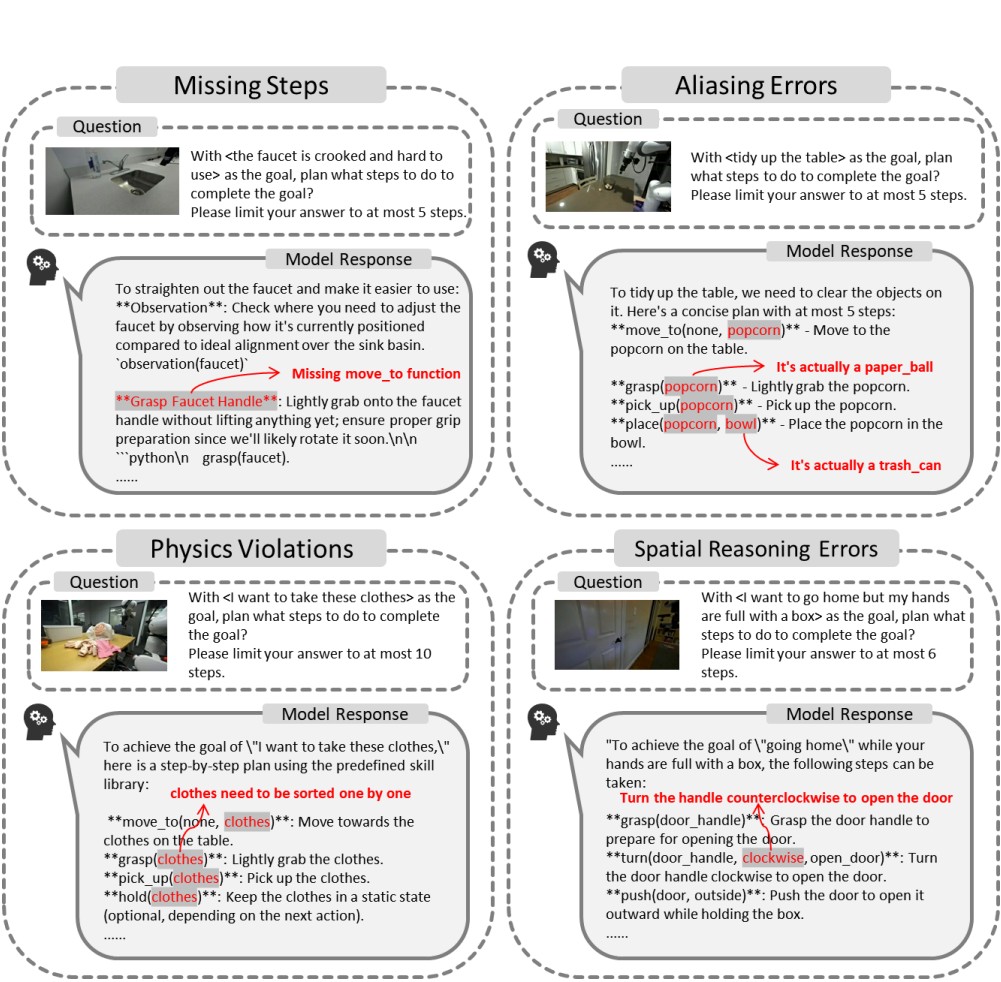

Figure 12: Representative error cases from our evaluation.

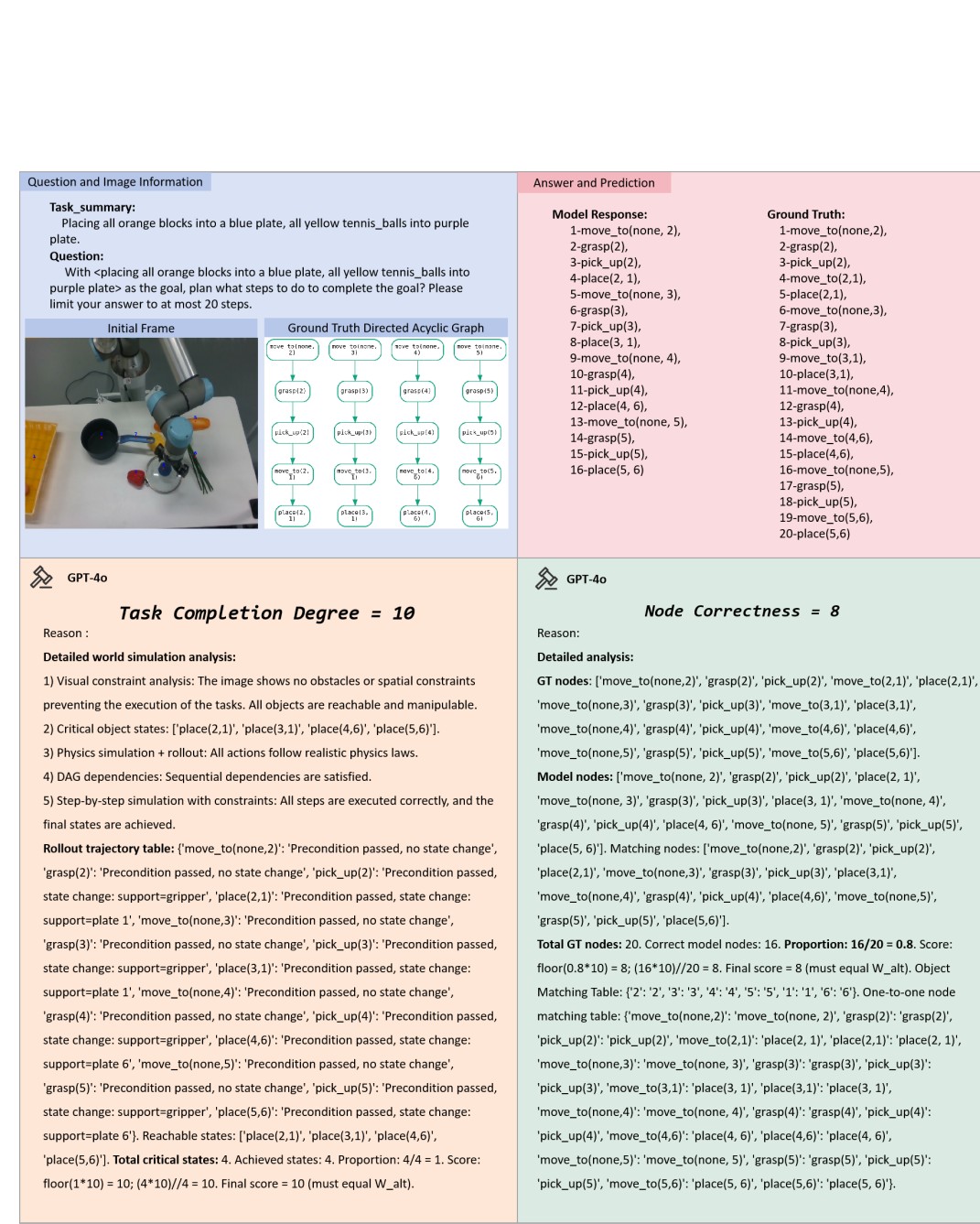

Figure 13: Successful evaluation examples illustrating robustness.

---

### Prompt: Video → NL Task & Steps (Single-Arm)

You will analyze a video (represented by image frames) of a robotic arm performing a specific task, where the task is described as: {desc}. Note that the referenced task summary might not be accurate or complete. Your task is to identify the primary task during the video with the help of the referenced description, summarize the task and rewrite the description, extract the necessary steps to complete it, and specify the frame range for each step. Follow these instructions:

1. **Task Identification**: First, identify the main task the robotic arm is performing. This task could be a clear goal or a series of related activities (e.g., assembling furniture, repairing equipment, preparing food, etc.). Briefly describe the primary task in one sentence.

2. **Step Extraction**: Once the task is identified, extract the key steps required to complete it, ensuring that each step is clearly described and logically ordered. Each step may include:
   - Specific actions (e.g., tightening screws, stirring mixtures, pressing buttons, etc.)
   - Frame window: Specify the start and end frame for each step (from 0 to {maxframeid}, since the video has {cnt} frames).

3. **Frame Range Constraints**:
   - **No Overlapping Frames**: Ensure that the frame ranges of each step do not overlap with each other. Each frame should be assigned to exactly one step.
   - **Full Frame Coverage**: Ensure that all {cnt} frames (from 0 to {maxframeid}) are included in the steps. No frames should be missed or duplicated.
   - **Must start from frame 0**

4. **Notes**:
   - Please annotate as finely as possible and try not to have more than ten frames of the same thing being done (unless it is difficult to distinguish).
   - A step has only one verb (unless two or more actions are strictly performed simultaneously).
   - Please ensure the accuracy of labeling and image matching.
   - For example, if the gripper changes from an open state to a closed state in a sequence of frames, the action is called `pick`; On the contrary, if it changes from a closed state to an open state, the action is called `release`.

5. **Failure Identification**: If the robotic arm attempts an action but does not succeed, clearly indicate this in the step description. For example, if the robotic arm tries to pick up a block but fails, the step description should be something like `Attempt to pick up a block but fails`.

6. **Output Format**: Provide the task description and steps in two parts, formatted as JSON:
   - **Task Summary**: A string summarizing the primary task in the video without mentioning the subjects - the robotic arm.
   - **Steps**: An array where each element represents a step, containing:
     - `step_description`: A concise description of the step with the action being performed in the format of verb phrases without mentioning the subjects - the robotic arm (e.g., "Add syrup in the glass").
     - `start_frame`: The start frame of the step (from 0 to {cnt-1}).
     - `end_frame`: The end frame of the step (from 0 to {cnt-1}).

**Task Description**: {desc}

\*\*Example Output Format 1: (This is an example with a total frame length of 30, and the specific situation depends on the actual frame length.)\*\*

```json
{{
  "task_summary": "Assembling an office desk.",
  "steps": [
    {{
      "step_description": "Remove all components and screws from the package.",
      "start_frame": 0,
      "end_frame": 4
    }},
    {{
      "step_description": "Use a screwdriver to attach the legs to the tabletop.",
      "start_frame": 5,
      "end_frame": 14
    }},
    ...
    {{
      "step_description": "Ensure all screws are tight and the desk is stable.",
      "start_frame": 25,
      "end_frame": 29
    }}
  ]
}}
```

\*\*Example Output Format 2: ...

---

Figure 14: Prompt for converting single-arm robot videos into task and step descriptions in natural language.

---

### Prompt: Video → NL Task & Steps (Dual-Arm)

You will analyze a video (represented by image frames) of a dual-arm robotic system performing a specific task, where the task is described as: {desc}. Note that the referenced task summary might not be accurate or complete. Your task is to identify the primary task during the video with the help of the referenced description, summarize the task and rewrite the description, extract the necessary steps to complete it, and specify the frame range for each step. Follow these instructions:

1. **Left-Right Hand Identification**: The video is recorded from a first-person view of the dual-arm robotic system. Your first task is to accurately identify which hand is the left arm (`[left]`) and which hand is the right arm (`[right]`). This is crucial as you proceed with the task analysis. Use visual cues such as the relative position of the hands, their orientation, and any distinguishing features to determine which hand is on the left and which is on the right.

2. **Task Identification**: Once the left and right arms are correctly identified, determine the main task the robotic system is performing. This task could be a clear goal or a series of related activities (e.g., assembling furniture, repairing equipment, preparing food, etc.). Briefly describe the primary task in one sentence.

3. **Step Extraction**: After identifying the task and distinguishing the left and right arms, extract the key steps required to complete the task, ensuring that each step is clearly described and logically ordered. **Make sure the following criteria are met**:
   - The **first step** must always start with `start_frame` equal to `0`, and the **last step** must end with `end_frame` equal to {cnt-1}.
   - Every step must explicitly describe the actions of both the left arm (`[left]`) and right arm (`[right]`). If one arm is inactive during a step, mention its inactive state (e.g., `[left]` holds the object steady while `[right]` tightens the screw). If both arms are involved in the same action, use `[both]` to describe their joint activity, but still ensure to detail the left and right arm roles (e.g., `[both]` lift the object, with `[left]` supporting the base and `[right]` holding the top).
   - **Frame window**: Specify the start and end frame for each step (from `0` to {cnt-1}, since the video has {cnt} frames).
   - **Handling Camera Obstruction**: If at any point the view is obstructed by one or both of the robotic arms, add a special marker `[block]` at the beginning of the step description. Then, based on the context before and after the obstruction, infer the likely actions taking place and provide the most accurate analysis possible. The `[block]` token should be used only when the camera is blocked, and after the token, describe the inferred task as usual (e.g., `[block]` `[left]` holds the object steady, `[right]` tightens the bolt).

4. **Frame Range Constraints**:
   - **No Overlapping Frames**: Ensure that the frame ranges of each step do not overlap with each other. Each frame should be assigned to exactly one step.
   - **Full Frame Coverage**: Ensure that all {cnt} frames (from `0` to {cnt-1}) are included in the steps. No frames should be missed or duplicated.

5. **Notes**:
   - Please annotate as finely as possible and try not to have more than ten frames of the same thing being done (unless it is difficult to distinguish).
   - A step has only one verb (unless two or more actions are strictly performed simultaneously).
   - Please ensure the accuracy of labeling and image matching.
   - For example, if the gripper changes from an open state to a closed state in a sequence of frames, the action is called `pick`; On the contrary, if it changes from a closed state to an open state, the action is called `release`.

6. **Output Format**: Provide the task description and steps in two parts, formatted as JSON:
   - **Task Summary**: A string summarizing the primary task in the video without mentioning the subjects - the robotic arms.
   - **Steps**: An array where each element represents a step, containing:
     - `step_description`: A concise description of the step, specifying the actions of `[left]`, `[right]`, or `[both]` arms (e.g., `[left]` holds the frame, `[right]` tightens screws, `[both]` lift the object). Always describe `[left]` first and `[right]` second, when applicable.
     - `start_frame`: The start frame of the step (from `0` to {cnt-1}).
     - `end_frame`: The end frame of the step (from `0` to {cnt-1}).

**Task Description**: {desc}

**Example Output Format 1: (This is an example with a total frame length of 30, and the specific situation depends on the actual frame length.)**

```json
{
  "task_summary": "Assembling an office desk.",
  "steps": [
    {
      "step_description": "[left] removes all components from the package
              while [right] holds the package steady.",
      "start_frame": 0,
      "end_frame": 4
    },
    ...
    {
      "step_description": "[both] ensure all screws are tight and the desk is stable.",
      "start_frame": 29,
      "end_frame": 29
    }
  ]
}
```

**Example Output Format 2: ...

Figure 15: Prompt for converting dual-arm robot videos into task and step descriptions in natural language.

**Prompt: Video → NL Task & Steps (Mobile Manipulator)**

You will analyze a **first-person perspective (ego-centric) video** (represented by image frames) of a robot performing a specific task, where the task is described as: "{desc}". Note that the referenced task summary might not be accurate or complete.

Your task is to identify the primary task during the video with the help of the referenced description, summarize the task and rewrite the description, extract the necessary steps to complete it, and specify the frame range for each step. Each step will include a **state** and a corresponding **action description**. Please ensure that the division of atomic tasks is as detailed as possible, the task description is as clear as possible, and there are no vague descriptions.

1. **Task Identification**: First, identify the main task the robot is performing. This task could be a clear goal or a series of related activities (e.g., assembling furniture, repairing equipment, preparing food, etc.). Briefly describe the primary task in one sentence.

2. **Step Extraction**: Once the task is identified, extract the key steps required to complete it, ensuring that each step is clearly described and logically ordered. Each step consists of two parts:

- **State**: The state of the robot during the step—whether it is moving (`[mobile]`), manipulating (`[manipulation]`), or observing (`[observation]`):

    - `[mobile]`: The robot changes position, but the camera view does not rotate. Describe the movement of the robot (e.g., "Move towards the table").
    - `[manipulation]`: The robot's position and camera view remain static, but the hands are performing actions. Use `[left]`, `[right]`, or `[both]` to describe the actions of the hands (e.g., "`[left]` holds the frame, `[right]` tightens the screws").
    - `[observation]`: This state **must** be triggered when the ego-centric camera's view or orientation changes, indicating the robot is observing or searching for an object. This step is required as a transition whenever the camera view changes, regardless of the robot's other actions. For each `[observation]` state, use the following format:
        * **[goal]**: Clearly specify the object or scene the robot is searching for, using `[find]` to indicate what the robot is looking for or detecting (e.g., "`[find]` the legs of the desk").
        * **[current object or scene]**: Describe what the robot currently sees (e.g., "A table and a chair are visible").
        * **[search result]**: Indicate whether the target object has been found (`yes`, `no`, `part`).

- **Action Description**: A concise description of what happens during the step, based on the state.

3. **Output Format**: Provide the task description and steps in two parts, formatted as JSON:

- **Task Summary**: A string summarizing the primary task in the video without mentioning the subjects - the robot's hands or the mobile base.

- **Steps**: An array where each element represents a step, containing:

    - state: The current state of the robot (`[mobile]`, `[manipulation]`, or `[observation]`).
    - action_description: A detailed description of the action in the current state (e.g., "`[mobile]` Move towards the table", "`[manipulation]` `[left]` holds the frame, `[right]` tightens the screws", "`[observation]` `[goal]`: `[find]` the legs of the desk. `[current object or scene]`: A table is visible. `[search result]`: no"). Always describe `[left]` first and `[right]` second if the two hands have different functions, when applicable.
    - start_frame: The start frame of the step (from 0 to {cnt-1}).
    - end_frame: The end frame of the step (from 0 to {cnt-1}).

4. **Frame Range Constraints**:

- **No Overlapping Frames**: Ensure that the frame ranges of each step do not overlap with each other. Each frame should be assigned to exactly one step.

- **Full Frame Coverage**: Ensure that all {cnt} frames (from 0 to {cnt-1}) are included in the steps. No frames should be missed or duplicated.

5. **Important Notes**:

- Please annotate as finely as possible and try not to have one step with more than ten frames doing the same thing (unless it is surely difficult to distinguish).

- A step has only one verb (unless two or more actions are strictly performed simultaneously).

- Please ensure the accuracy of labeling and image matching.

- The video consists of exactly {cnt} frames. Therefore, ensure that the **end frame** for the last step is always {cnt-1}, and no frame should exceed this value.

- Ensure that the **[observation]** state appears **at the beginning** of the task to search for the initial target object, and as needed thereafter.

- For **[observation]** states, use the special tokens **[goal]**, **[current object or scene]**, and **[search result]**.

- Always ensure **[observation]** concludes with `search result: yes` before transitioning to `[mobile]` or `[manipulation]`.

- **Left and Right Hand Determination**: Use global context, spatial relationships, and object positions to distinguish between `[left]` and `[right]` hands.

**Task Description**: {desc}

**Example Output Format (for a video with 60 frames):**

```json
{
    "task_summary": "Assembling an office desk.",
    "steps": [
        {
            "state": "[observation]",
            "action_description": "[goal]: [find] the legs of the desk. [current object or scene]:
                A table is visible. [search result]: no.",
            "start_frame": 0,
            "end_frame": 1
        },
        ...
    ]
}
```

Figure 16: Prompt for converting mobile-manipulator robot videos into task and step descriptions in natural language.

---

**Manipulation Function List**

**Functions for the actions of a gripper:**

- `move_to(object, target_object)`: Move the gripper. First parameter is the object currently held (`none` if empty), second parameter is the target object. Example: `move_to(none, towel)`, `move_to(panda_toy, bowl)`.
- `hold(object)`: Keep an object in static hold. Not applicable if gripper is empty. Example: `hold(cup)`.

**Functions for grabbing and releasing:**

- `pick_up(object)`: Pick up a graspable object. Example: `pick_up(apple)`.
- `grasp(object)`: Lightly hold an object (pick-upable or not). Example: `grasp(door_handle)`.
- `place(object, target_object)`: Place an object at a location or relative position. Example: `place(apple, table)`, `place(apple, right_of_banana)`.

**Functions for using a tool:**

- `scoop(tool, contents, container)`: Use a tool to scoop contents. Use `unknown` if contents uncertain. Example: `scoop(spoon, water, bowl)`.
- `pour(container, contents, target_container)`: Pour contents into target. Example: `pour(bowl, water, pot)`.
- `wipe(tool, object, target_object)`: Wipe object using tool on target. Example: `wipe(towel, water, table)`.
- `stir(tool, contents, target_container)`: Stir contents with tool in container. Example: `stir(spoon, soup, pot)`.
- `draw(tool, character, target_object)`: Draw a character using a tool on target. Example: `draw(marker, 'A', whiteboard)`.
- `cut(tool, object, target_object)`: Cut an object with a tool on target. Example: `cut(knife, tomato, chopping_board)`.

**Functions for interacting directly:**

- `fold(object, target_position)`, `unfold(object, target_position)`: Fold or unfold object to target position.
- `turn(object, direction, state_of_target_object)`: Rotate object to target state. Directions: {clockwise, anticlockwise, up, down, forward, backward, left, right}. Example: `turn(faucet, clockwise, middle_of_sink)`.
- `press(tool, object)`: Press object using tool (`none` if no tool). Example: `press(none, red_button)`.
- `push(object, target_location)`, `pull(object, target_location)`: Push or pull object to target. Example: `push(chair, under_of_table)`, `pull(towel, right_side_of_table)`.
- `insert(object, target_object)`, `pullout(object, target_object)`: Insert or remove object from target. Example: `insert(plug, socket)`, `pullout(plug, socket)`.

**Dual-arm specific:**

- `transfer(left/right, right/left, object)`: Transfer object between hands. Example: `transfer(left, right, bottle)`.

**Mobile-manipulation specific:**

- `observation(object)`: Object not visible, needs to be found. Example: `observation(chair)`.
- `mobile(target_object)`: Object visible but distant, move robot to approach. Example: `mobile(table)`.

**No Operation:**

- `no_ops`: Stay still or maintain current state.

Figure 17: Predefined Manipulation function list used in NL-to-function conversion.

---

**Navigation Function List**

**Navigation:**

- `turn_left()`: Rotate the robot $90°$ to the left in place.
- `turn_right()`: Rotate the robot $90°$ to the right in place.
- `turn_back()`: Rotate the robot $180°$ in place (turn around).
- `go_forward(target_location)`: Move forward in the current facing direction until reaching the specified `target_location`. The parameter corresponds to the semantic name of the target location, e.g., `go_forward(dining_table)` or `go_forward(kitchen_door)`.

Figure 18: Predefined navigation function list used in natural-language-to-function conversion. The design ensures a compact yet expressive action space that supports systematic evaluation of compositional navigation instructions.

## Prompt: NL Steps → Function Sequence

**Task Description**
You need to process a set of **natural language step descriptions** for a robotic manipulation task and extract **generalized functions**. All generated functions must be selected from a predefined list of functions, ensuring that the function names and parameters match the definitions in the list. If the input action cannot be matched to any function in the list, use `special_action()` and generate a suitable action.

**Rules & Requirements**

1. **Select from Predefined Functions**:
   - All generated functions must be selected from the following categories:
     - **Object Manipulation**
     - **Movement and Navigation**
     - **Opening and Closing**
     - **Special Operations**
     - **no_ops**
     - `special_action()`
   - If the input action cannot be matched to any function in the list, use `special_action()`.
   - If the input natural language action description indicates a left-handed action and a right-handed action respectively, you need to match a function for each of the left-handed and right-handed actions. Note that the description with `[both]` tag means both hands operate simultaneously and it is also suitable to match function to each of them.

2. **Function Selection Rules**:
   - Select the most appropriate function based on the semantics of the action.
   - Refer to the explanations provided for each function to ensure correct selection.

3. **Output Format**:
   - The output must be in a single-line JSON array format.
   - Each function should be enclosed in quotes and separated by commas.
   - The order of functions should match the input.
   - The number of output functions must be strictly equal to the number of input **natural language step descriptions**.
   - If `special_action()` is used, include the generated description in the JSON.

**Predefined Functions List with Explanations**
**[Predefined Functions List with Explanations]...**
**Example List**
**Input:**
pick up cup, move to shelf with block, push ball, place book on desk, hover over table, [left] approach glasses case, [right] approach lid, [left] hold case steady, [right] move lid, [both] close laptop lid
**Output:**

```
["pick_up(object)", "move_to(object, target_object)", "push(object, target_object)",
 "place(object, target_object)", "no_ops",
 "left:move_to(object, target_object), right:move_to(object, target_object)",
 "left:no_ops, right:move_to(object, target_object)",
 "left:close(object), right:close(object)"]
```

**Task**
Convert the following natural language step descriptions into structured generalized functions:

Figure 19: Prompt for converting natural language steps into a sequence of predefined functions.

1404
1405
1406
1407
1408
1409
1410
1411
1412
1413
1414
1415
1416
1417
1418
1419
1420
1421
1422
1423
1424
1425
1426
1427
1428
1429
1430
1431
1432
1433
1434
1435
1436
1437
1438
1439
1440
1441
1442
1443
1444
1445
1446
1447
1448
1449
1450
1451
1452
1453
1454
1455
1456
1457

---

## Prompt: Function Instantiation

**Task Description:** Your task is to process a set of natural language action descriptions and generate function calls based on provided function templates. The function calls must accurately reflect the semantics of the input actions and use the correct objects or targets from the descriptions.

**Rules & Requirements**

- **Input:**
  - A list of natural language action descriptions (e.g., "reach for the juice bottle") and a list of function templates (e.g., "move_to(object)").
  - The number of function templates matches the number of action descriptions, and they correspond in order.
- **Output:**
  - A list of function calls that match the input actions and templates (e.g., "move_to(juice_bottle)").
  - The number of function calls must match the number of input descriptions.
  - If an object or target is composed of multiple words, separate them with underscores (_).
- **Function Call Rules:**
  - Replace the placeholder (e.g., object, target_object, goal, current_scene_or_object) in the function template with the appropriate object or target from the natural language description.
  - Ensure the function call accurately reflects the action described in the input.
- **Output Format:** The output must be a list of function calls in the correct format.

**Function Templates contain three types of "observation", "mobile" and "manipulation". The specific templates and explanations are as follows:**
[Predefined Functions List with Explanations]
**Examples**
**Input:**

- **Natural Language Descriptions:** "[goal]: [find] a cloth [current object or scene]: cabinets, countertop, and floor are visible [search result]: no", "move towards the cloth", "[right] places the oil bottle back on the counter", "[right] reaches for the bowl of shrimp", "[goal]: [find] spatula [current object or scene]: pan with shrimp and spatula are visible [search result]: yes", "reach for the juice bottle", "grasp the juice bottle", "lift the juice bottle", "move the juice bottle towards the blue sticky note", "place the juice bottle on the blue sticky note", "release the juice bottle".

- **Function Templates:** "observation(goal, current_scene_or_object, result)", "mobile(target_object)", "left:no_ops, right:place(object, target_object)", "left:no_ops, right:move_to(object, target_object)", "move_to(object)", "pick_up(object)", "move_to(object, target_object)", "place(object, target_object)".

**Output:**

```
["observation(a_cloth, ['cabinets','countertop','floor'],no)", "mobile(cloth)",
"left:no_ops, right:place(oil_bottle, counter)",
"left:no_ops, right:move_to(none, bowl_with_shrimp)",
"observation(spatula, ['pan_with_shrimp_and_spatula'], yes)",
"move_to(juice_bottle)", "pick_up(juice_bottle)",
"pick_up(juice_bottle)", "move_to(juice_bottle, blue_sticky_note)",
"place(juice_bottle, blue_sticky_note)",
"place(juice_bottle, None)"]
```

**Task** Convert the following natural language step descriptions and function templates to the special function calls. Language Descriptions:  Function Templates:

---

Figure 20: Prompt for instantiating predefined functions with specific object references.

---

**Prompt: Explicit → Implicit**

**Task:** Generate implicit task instructions for embodied robots based on a provided explicit task and (if available) an image. The goal is to produce natural, everyday expressions that imply the need for the target task without explicitly mentioning the target object or directly stating the task.

**Key Requirements:**

1. **Target Entity Focus:**
   - Identify the unique characteristic of the target entity from the explicit task and ensure that the implicit instruction reflects this characteristic.
   - The implicit instruction must have a direct association with the target entity and task, not abstract references.

2. **Everyday Scenarios and Language:**
   - Use casual, real-life scenarios to imply the need for the task.
   - Avoid technical terms or abstract expressions. The instruction should feel like a natural request from a human in daily life.

3. **Image Integration (if provided):**
   - Analyze the provided image directly to extract relevant information about the scene.
   - The image may contain multiple objects, including distracting household items unrelated to the task. Focus on the target object from the explicit task while ensuring the implicit instruction remains relevant and natural.
   - Do not rely on textual descriptions of the image; all visual information must come from analyzing the provided image itself.

4. **No Direct Mentions:**
   - Do not directly mention the target object or the explicit task.
   - The need for the task should be implied through observations, feelings, or everyday needs (e.g., "I feel parched," instead of "pour water into the cup").

5. **Output Format:**
   - Provide 5 implicit task instruction suggestions for each input.
   - The output must be in list format, with each instruction as a separate list item to ensure consistency and ease of post-processing.

**Examples:**
**Target object:** pouring liquid from a bottle into a cup **Provided Image:** A table with a bottle, an empty cup, and other unrelated items such as soap, a photo frame, and a gift bag.
**Instruction Output:**
   - "Looking at the empty cup on the table makes me realize how thirsty I am. Could you help me with that?"
   - …

**Target object:** watering a plant **Provided Image:** A windowsill with several potted plants, one with drooping leaves.
**Instruction Output:**
   - "This plant's leaves look a bit droopy today. Could you help bring it back to life?"
   - …

**Target object:** selecting a Batman toy **Provided Image:** A shelf filled with various superhero toys, including a Batman figure.
**Instruction Output:**
   - "I'm a big fan of DC series, please help me choose a suitable toy."
   - …

**Input Format:**
   - **Target object:** {task_summary}
   - **Provided Image:** {image_path}

**Output Format:**
   - **Instruction Output:**
     - "Instruction 1"
     - "Instruction 2"
     - "Instruction 3"
     - "Instruction 4"
     - "Instruction 5"

Figure 21: Prompt for converting explicit task instructions into implicit form.

---

**Prompt: Step-level Object Extraction**

The following is a long-term robot task instruction text. Please analyze the objects being manipulated and other objects mentioned contained in it and generate the output part in a strict format. All words are in the singular form. Object properties (if there are properties other than quantity) are placed in parentheses. You also need to analyze, what kind of object is this manipulation for, choosing from rigid (like bowl), articulated (like microwaves), deformable (like cloth), special (like liquid).

For example,

- task instruction: picking grapes and placing them on a plate; output: [manipulated: grape=rigid][other: plate]

- task instruction: picking up a bowl; output: [manipulated: bowl=rigid][other: none]

- task instruction: pick up a small coca-cola can and place it on a blue paper; output: [manipulated: can(small, coca-cola)=rigid][other: paper(blue)]

- task instruction: pouring dice from a white cup into a pink cup; output: [manipulated: dice=special, cup(white)=rigid][other: cup(pink)]

- task instruction: pouring water from one cup to another; output: [manipulated: water=special, cup=rigid][other: cup]

- task instruction: transferring a usb flash drive between two devices; output: [manipulated: usb flash drive=rigid][other: device]

- task instruction: folding a cloth; output: [manipulated: cloth=deformable][other: none]

- task instruction: opening a laptop; output: [manipulated: laptop=articulated][other: none]

- task instruction: closing a cabinet door; output: [manipulated: door(cabinet)=articulated][other: none]

- task instruction: wiping a table; output: [manipulated: cloth=deformable][other: table]

- task instruction: writing the number 14 on a whiteboard; output: [manipulated: mark pen=rigid][other: whiteboard]

Figure 22: Prompt used to extract **objects** at the step level from video frames.

---

**Prompt: Step-level Action Extraction**

Below are task instructions for long-term robotic tasks. Your goal is to extract the primary action from each task instruction.

For instance:

- Task instruction: dragging a strainer backwards, should return: `drag`

- Task instruction: Spelling "THU" with blocks, should return: `spell`

- Task instruction: inserting a three-pronged object into its matching slot, should return: `insert`

- Task instruction: filling a bottle with water from a dispenser, should return: `fill`

- Task instruction: pick apple and place in the drawer, should return: `pick, place in`

These examples illustrate the type of concise action extraction required. The task instruction provides a high-level overview of the task, and your job is to distill it into its core action(s). This prompt is designed for the Gemini model, and it is crucial to focus on identifying the main action verbs that define the task.

Figure 23: Prompt used to extract **actions** at the step level from video frames.

---

**Prompt: Step-level Scene Label Extraction**

Please analyze the following image scene and provide a set of scene tags. The specific example of the tags is as follows:

- {'Primary Tag': 'Lab Scene', 'Secondary Tag': 'Robotics Testing Area', 'Tertiary Tag': 'Shape-Sorter Toy', 'Perspective Tag': 'Third-person View'}

- {'Primary Tag': 'Lab Scene', 'Secondary Tag': 'Robot control room', 'Tertiary Tag': 'Workbench', 'Perspective Tag': 'Third-person View'}

- {'Primary Tag': 'Family Scene', 'Secondary Tag': 'Kitchen', 'Tertiary Tag': 'Chopping Board', 'Perspective Tag': 'First-person View'}

Among them, the tertiary tag represents the specific background of these images (for example, all operations in the experiment are carried out on the chopping board); the secondary tag is a collection of tertiary tags, such as stoves, chopping boards, these tertiary scene tags should be clustered into the kitchen, this secondary scene tag, basket frames, and basketball hoops, these tertiary scene tags should be clustered into the basketball court, this secondary scene tag; the primary tag is the clustering of secondary tags, such as kitchens, living rooms, and bedrooms, these secondary scene tags should be clustered into the family scene, this primary tag; basketball courts, volleyball courts, and ping-pong tables, these secondary scene tags should be clustered into the gym, this primary tag; the perspective tag indicates whether the image is shot from a first-person or third-person perspective. The first-person perspective should have the entire scene moving with the operation of the robotic arm, while in the third-person perspective, the overall scene should remain stationary, with only the robotic hand moving.

**Note:**

1. If you don't know what scene tags to fill in, you can fill in ¡unknown¿ as the answer.

2. The Primary Tag, Secondary Tag, and Tertiary Tag must fill in the background information of the several frames of images I gave you. Please do not fill in other information!!!

**Again, your result must use the following format:**

- Before providing your tag answers, please explain the reasoning behind the labels you have given.

```
[
    {
        "Primary Tag": "the primary scene tag",
        "Secondary Tag": "the secondary scene tag",
        'Tertiary Tag': "the tertiary scene tag",
        "Perspective Tag": "the perspective type of images"
    }
]
```

Figure 24: Prompt used to extract **scene labels** at the step level from video frames.

---

**Prompt: Functional Attribute QA Generation**

As an AI visual assistant, you are tasked with analyzing an image that includes a single marked bounding box (colored green) and generate a multi-choice question-answer pair. The bounding box format is [x, y, width, height], x and y typically represent the coordinates of the top-left corner of the bounding box, while width and height represent the width and height of the box. The image bounding box: {bounding_box}. Initially, you must acquire a comprehensive and intricate understanding of the image, ideally down to each pixel area. Your primary focus should be on deciphering the specific objects or contexts associated with the marks I have placed within the image, as this is of paramount importance. Your analysis will encompass two principal functions:

**Bounding Box Description**: For each marked bounding box in the image, provide a thorough description using natural language. Detail attributes such as the category, type, color, functionality, and other characteristics of the object, including its location, condition, and any additional pertinent attributes. Envision yourself observing directly and convey your observations as thoroughly and promptly as possible.

*– Template for Role 1*:
*'Bounding Box 1': [Comprehensive Description]*
*Continue in this manner for marked bounding box.*

**Multi-choice Question-Answer Pair Generation**: Using the functionality that you describe in Role 1, generate a question about the functionality or use of the object inside the Bounding Box. Provide multiple-choice options and the correct answer. Ensure the question and options are relevant to the object's functionality. The choice in the question must be a little bit difficult to answer, make sure is a good question to distinguish where the model are smart or not.

*– Template for Role 2*:
*"Question": What functionality does the object inside the Green Bounding Box have? (A) ... (B) ... (C) ... (D) ...*
*"Answer": (...)*
*–Example*:
*"Question": What functionality does the object inside the Green Bounding Box have? (A) Used to store water (B) Used to store food (C) Used for drinking hot beverages (D) Used to plant flowers*
*"Answer": (C)*

Figure 25: Prompt for generating question-answer pairs describing object functionalities.

---

**Prompt: Q1 Evaluation - Action List Extraction**

**Task:** You are given an input dataset containing a robotic manipulation task goal, a previously executed step, and a response describing the remaining steps. Your task is to extract structured action plans in a specific function format.

**Instructions:**

1. **Extract Key Information:** Identify the task goal from the `prompt` field and assign it to the `"task_summary"` field. Extract action functions from the `previous_step` and `response` fields to construct the sequence of necessary steps in `"plan_step"`.

2. **Strict Action Function Format:** Use only the predefined action functions listed below. Do not modify function names or introduce new ones. Ensure all function names match exactly. Arguments (`object`, `target_object`, `carry_object`, `direction`) should be generalized but faithful to the task.

3. **Maintain Execution Order:** The `"plan_step"` list should follow the correct execution order.

4. **Reasoning Explanation:** Provide a `"reason"` field explaining how the `"task_summary"` and `"plan_step"` were derived, including how you determined the format (single-arm vs dual-arm).

**Predefined Action Functions:** `""" + PREDEFINED_ACTIONS + """`

**Output Format (JSON):**
*For Single-Arm Tasks:*

```
{
  "task_summary": "<task goal>",
  "plan_step": ["<action_function_1>", "<action_function_2>", ...],
  "reason": "<your reasoning>"
}
```

*For Dual-Arm Tasks:*

```
{
  "task_summary": "<task goal>",
  "plan_step": [
    "<action_function_1>",
    "left:<action_function_2>, right:<action_function_3>",
    ...
  ],
  "reason": "<your reasoning>"
}
```

The data provided is as follows: {`data`}. Please output your results as required.

Figure 26: Prompt for extracting structured action lists from model outputs for Q1-type questions.

---

**Prompt: Q1 Evaluation - Scoring**

**Task:** Act as a judge of embodied multimodal task planning. Evaluate and score the planning capabilities of multimodal models in embodied intelligence scenarios.

**Input & Data Format:** You receive:

- **Image**: Current scene with visual constraints (obstacles, spatial layouts, object positions)
- **GT action list**: Ground Truth actions from Gemini video segmentation
- **GT DAG**: Task dependencies (parallel/sequential relationships)
- **Model plan**: Action list output by the evaluated model

Format: {{'GT action list': [...], 'GT dag': [...], 'model plan action list': [...]}}

**Mode Detection:**

- **Standard Mode**: Textual object names (e.g., `pick_up(apple)`)
- **CSS Mode**: Numeric object IDs (e.g., `pick_up(3)`)

**Predefined Action Functions:** `""" + PREDEFINED_ACTIONS + """`

**Evaluation Criteria (0–10 points each):**

1. **Node Correctness**: Match skill-object-parameter nodes between GT and model outputs. *Scoring:* floor((correct/total) * 10). Flexible matching in Standard Mode; strict in CSS Mode.

2. **Task Completion Degree**: Count achieved critical object state changes (turn, place, pick_up, push/pull). Ignore robot motions (e.g., move_to, grasp). *Scoring:* floor((achieved/total) * 10).

**Flexible Equivalence (Standard Mode Only):** Functional equivalence (e.g., drawer $\sim$ drawer_handle, lamp_switch $\sim$ power_button) is accepted.

**Output Format (JSON):**

```
{
    "node_correctness": {
        "reason": "... GT vs Model nodes, matching table, formulas ...",
        "result": x
    },
    "task_completion_degree": {
        "reason": "... visual constraints, rollout simulation ...",
        "result": y
    },
    "planning_issue_analysis": {
        "issue_types": [...],
        "detailed_analysis": "..."
    },
    "comprehensive_evaluation": "..."
}
```

**Final Guidelines:** Enforce mode detection, sequential rollout with DAG constraints, strict vs flexible matching rules, state-tracking for critical object changes, and consistent scoring formulas. Include required artifacts in reasoning: object matching tables, rollout trajectories, and issue categorization.

Figure 27: Prompt for scoring model outputs for Q1-type questions based on ground truth.

---

**Prompt: Q2 Evaluation - Action Extraction**

**Task:** Extract the next step from the given response and represent it in a structured robotic action format.

**Action Format:** The extracted step must follow the format: `skill(element1, element2, ...)`

**Examples:**

- `grasp(microwave_handle)`
- `push(microwave_handle, close)`
- `move_to(none, drawer)`

**Input:** Response: {response}

**Output Instruction:** Return **only** the extracted step in the correct format. Do not include any explanations, extra text, or additional symbols.

Figure 28: Prompt for extracting structured actions from model outputs for Q2-type questions.

---

**Prompt: Q2 Evaluation - Prompt-based Scoring**

**Task:** You are given two robot action steps (an *extracted* step from a model and a *ground-truth* step). Evaluate their similarity using the prescribed, quantitative criteria below and provide a concise justification for each sub-score.

**Input Data Format:**

```
{
  "extracted_step": "The step extracted from model response",
  "gt_step": "The ground truth step"
}
```

**Evaluation Criteria:**

1. **Skill usage accuracy (0 or 1 point).** Consider only the skill/action token (e.g., `grasp`, `push`, `move_to`) in both steps. Award 1 iff the skills are exactly identical (strict match after normalization); otherwise award 0.
   *Normalization rule:* lowercase, strip extra spaces/underscores for comparison (e.g., `Pick_Up` → `pickup` for matching purposes).

2. **Operation object reasonableness (0, 0.5, or 1 point).** Compare the object argument(s) referenced by the operations:
   - `1.0`: Identical or clearly the same referent (alias/orthographic variants accepted, e.g., `door_handle` vs `door handle`).
   - `0.5`: Related or part/whole/category-level match (e.g., `table` vs `table_leg`, `cup` vs `mug`).
   - `0.0`: Unrelated or incompatible objects.

   Evaluate object similarity using semantic equivalence and task context.

3. **Parameter accuracy (0, 0.5, or 1 point).** Evaluate additional parameters (e.g., target positions, directions, contents). **Important:** if Skill score = 0 or Object score = 0, Parameter score = 0. Otherwise:
   - `1.0`: Parameters fully correct and precise for execution.
   - `0.5`: Parameters partially correct or imprecise but salvageable.
   - `0.0`: Parameters incorrect or irrelevant.

**Evaluation Guidelines:**

- Enforce **strict** skill matching; be **flexible** on objects in Standard mode (use context), but adhere to strict ID matching in CSS/ID mode if applicable.
- Provide brief, explicit reasons for each sub-score (one-line justification).
- Consider execution precision required by robotic control when judging parameters (e.g., `push(drawer, close)` vs `push(drawer, slight_nudge)`).
- Examples:
  - Extracted: `push(drawer, close)` — GT: `push(drawer_handle, open)` → Skill=1, Object=0.5 (part-whole), Parameter=0.5 (partial).
  - Extracted: `move_to(none, table)` — GT: `move_to(none, table)` → Skill=1, Object=1, Parameter=1.

**Output Format (JSON):** Return a single JSON object strictly matching the structure below. Each '"reason"' entry should be a concise justification for the assigned score.

```
{
  "skill_usage_accuracy": {"result": x, "reason": "brief explanation of evaluation"},
  "operation_object_reasonableness": {"result": y, "reason": "brief explanation of evaluation"},
  "parameter_accuracy": {"result": z, "reason": "brief explanation of evaluation"}
}
```

The data provided is as follows:
Extracted step: $extracted_step$ Ground truth step: $gt_step$
Please output your results exactly in the JSON format above.

Figure 29: Prompt for scoring Q2-type questions based on model outputs and prompt instructions.

---

**Prompt: Q3 Evaluation - Yes/No Conversion**

**Task:** From a given response, extract only the binary decision: `"yes"` or `"no"`. No explanations, additional words, or formatting are allowed.

**Input Format:**

```
Response: {response}
```

**Output Requirement:**

- Return strictly one token: `"yes"` or `"no"`.
- Output must be lowercase.
- No punctuation, whitespace, or extra text is permitted.

**Examples:**

- Response: "Yes, that is correct." → Output: `yes`
- Response: "No, it does not work." → Output: `no`

Figure 30: Prompt for converting model outputs into yes/no answers for Q3-type questions.

