# OpenReview forum: "Robobench: A Comprehensive Evaluation Benchmark For Multimodal Large Language Models as Embodied Brain"
_ICLR.cc/2026/Conference — Submitted to ICLR 2026_

### Official Review · Reviewer_ci1f · 2025-10-19

**Soundness:** 3
**Presentation:** 2
**Contribution:** 2
**Rating:** 4
**Confidence:** 4

**Summary:**

The paper introduces RoboBench, a comprehensive benchmark designed to evaluate MLLMs as the cognitive "embodied brain" for robotic manipulation. RoboBench systematically evaluates MLLMs across five key dimensions: instruction comprehension, perception reasoning, generalized planning, affordance prediction, and failure analysis, spanning 25 tasks and over 6000 QA pairs. A core contribution, as stated by the authors, is its "MLLM-as-world-simulator" framework, which assesses long-horizon planning by simulating the embodied feasibility of predicted steps rather than just symbolic accuracy. Experiments on 14 state-of-the-art MLLMs revealed fundamental limitations, including difficulties with implicit instruction comprehension, spatiotemporal reasoning, and execution failure diagnosis.

**Strengths:**

+ The benchmark is deliberately scoped to the cognition layer of the manipulation pipeline, focusing on five interdependent capabilities: instruction comprehension, perception reasoning, generalized planning, affordance prediction, and failure analysis.
+ The benchmark ensures realism by curating datasets across diverse embodiments (single-arm, dual-arm, and mobile manipulation), objects with rich attributes, multi-view scenes with occlusion, and memory-driven navigation.
+ The paper provides a clear presentation and articulates a strong motivation for the benchmark, addressing the limitations of existing evaluations that either focus too narrowly on execution success or lack task realism.
+ The benchmark is comprehensive in its scale, covering 15 capabilities, 25 tasks, and a total of 6077 question-answer pairs.

**Weaknesses:**

+ The paper introduces an "MLLM-as-world-simulator" framework for planning evaluation, which functions as an "MLLM-as-judge". This approach is susceptible to well-documented biases [1, 2, 3] inherent in using LLMs for evaluation, such as positional bias or favoring models with similar architectures or response styles. The paper does not analyze or mitigate these potential biases, which may affect the objectivity of the planning scores.

+ The benchmark's evaluation of long-horizon planning relies on manually annotated DAGs to represent ground-truth dependencies. However, the paper does not report inter-annotator agreement (IAA), making it difficult to gauge the reliability of these crucial annotations.

+ The scope of the benchmark is limited to table-top manipulation tasks, which may represent relatively short and constrained interactions from the perspective of an MLLM's capabilities.

+ While a key novelty is the use of real-world data, the evaluated tasks are not fundamentally distinct in nature from those found in previous simulation-based benchmarks [4, 5]. As a result, the insights derived from the evaluation, while confirming existing knowledge, may not offer substantially new perspectives on the cognitive gaps of MLLMs in embodied AI.

+ The failure analysis dataset is constructed by synthetically perturbing correct plans to create planning errors (e.g., wrong object, missing step) due to the lack of real-world failure data. This synthetic generation may not capture the true distribution and complexity of failures encountered in real robotic execution, potentially limiting the practical value of the conclusions drawn from this benchmark dimension.

+ A minor presentation issue exists in Figure 1, where the top-right performance comparison radar chart is rendered at a low resolution, causing the text and plot details to be blurry.

[1] Panickssery, Arjun, Samuel Bowman, and Shi Feng. "Llm evaluators recognize and favor their own generations." Advances in Neural Information Processing Systems 37 (2024): 68772-68802.

[2] Chen, Guiming Hardy, et al. "Humans or llms as the judge? a study on judgement biases." arXiv preprint arXiv:2402.10669 (2024).

[3] Dorner, Florian E., Vivian Yvonne Nastl, and Moritz Hardt. "Limits to scalable evaluation at the frontier: LLM as judge won’t beat twice the data." The Thirteenth International Conference on Learning Representations.

[4] Li, Manling, et al. "Embodied agent interface: Benchmarking llms for embodied decision making." Advances in Neural Information Processing Systems 37 (2024): 100428-100534.

[5] Yang, Rui, et al. "Embodiedbench: Comprehensive benchmarking multi-modal large language models for vision-driven embodied agents." arXiv preprint arXiv:2502.09560 (2025).

**Questions:**

1. Will you provide the analysis or justify how you mitigate the potential biases when using MLLM-as-judge style metrics?

2. What are the inter-annotator agreement statistics and QA procedures for the DAG annotations?

3. What insights uniquely arise from your real-world setup vs simulator-based setup (e.g., any ablations showing differences)?

4. I know this question might be hard, but if possible, could you please validate the synthetic failure set against real robot logs (or in a simulation environment) in a small subset? It is okay to skip this question.

---

### Official Review · Reviewer_pDCv · 2025-10-31

**Soundness:** 3
**Presentation:** 4
**Contribution:** 1
**Rating:** 2
**Confidence:** 4

**Summary:**

This work proposes an embodiment benchmark that systematically evaluates embodied agents across multiple dimensions, including instruction comprehension, perceptual reasoning, generalized planning, affordance prediction, and failure analysis, using a dataset of 6,077 QA pairs. It introduces a task planning evaluation framework based on a work-simulation rollout process, which measures task completion rates according to a directed acyclic graph (DAG) of subtasks. Finally, the benchmark is used to evaluate both open-source and closed-source MLLMs, providing valuable insights derived from the comparative results.

**Strengths:**

The writing is simple and clear, resulting in good overall readability.

The benchmark effectively assesses long-horizon planning ability by simulating whether the generated plans achieve key object-state milestones.

It comprehensively covers diverse hardware configurations including bimanual, single-arm, and mobile robot setups as well as multiple task viewpoints, enhancing its generality and applicability.

**Weaknesses:**

The overall work feels engineering-oriented, focusing mainly on data curation rather than proposing new methods to improve MLLM performance, which limits the novelty and conceptual contribution of the paper.

The five key capabilities defined in this work appear to have some overlap. For example, embodied instruction comprehension could arguably be considered part of embodied generalized planning, as both involve understanding structured task sequences.

The motivation and practical value of the failure case analysis seem somewhat unconvincing. Moreover, as shown in Figure 2, identifying the actual cause of failure appears difficult without making additional assumptions.

Based on Figure 4, the so-called systematic evaluation seems somewhat biased — with relatively few instances of affordance reasoning and error analysis, while planning tasks dominate the benchmark.

**Questions:**

Would it be possible to perform a systematic evaluation of embodied agents across multiple dimensions by combining existing benchmarks, rather than creating an entirely new one?

Could you clarify the practical value of the error analysis QA? There are many different types of failure cases, and it seems difficult to anticipate or comprehensively cover them in advance.

Are all the QA pairs based on single images? If so, it may be challenging to accurately capture actions such as whether the robot is opening or closing a drawer.

It would be helpful to briefly discuss the cost and feasibility of conducting benchmarking experiments on RoboBench.

---

### Official Review · Reviewer_VnZb · 2025-10-31

**Soundness:** 2
**Presentation:** 2
**Contribution:** 2
**Rating:** 4
**Confidence:** 3

**Summary:**

This paper presents a benchmark for evaluating the cognitive capabilities of MLLMs as brains for robotic manipulation tasks, covering five dimensions—Instruction Comprehension, Perception Reasoning, Generalized Planning, Affordance Prediction, and Failure Analysis. The benchmark is constructed from a mix of large-scale real robotic datasets and curated in-house data, with an emphasis on realistic, diverse embodiments, object properties, and scene configurations. Notably, RoboBench introduces a planning evaluation pipeline using an MLLM-as-world-simulator framework to measure plan feasibility and long-horizon reasoning. Evaluation results on 14 MLLMs reveal limitations of current models in implicit goal inference, perception, planning robustness, affordance reasoning, and diagnosis of execution failures.

**Strengths:**

1. The inclusion of the multi-view planning and error analysis task are good, which complements some existing evaluations for MLLM in embodied tasks.

**Weaknesses:**

1. The paper mostly uses existing data from other benchmarks to construct this evaluation, without any notable principled data curation method. The goal seems to be mostly just extending coverage of existing benchmarks and meshing together aspects from previous evals. The engineering effort is useful, yet research contributions are limited.

2. Comparison with relevant baseline benchmarks in Table 1 seems to be subjective and somewhat dubious. For example, it is unclear what is being referred to as Robustness Evaluation and why RoboBench has this while others do not.

**Questions:**

Additional citations and comparisons: ECBench, ManipBench, BEAR

---

### Official Review · Reviewer_JASz · 2025-11-01

**Soundness:** 3
**Presentation:** 3
**Contribution:** 2
**Rating:** 4
**Confidence:** 4

**Summary:**

This paper introduces a comprehensive benchmark that evaluates MLLMs as the embodied brain for robotic manipulation. It covers five core cognitive dimensions: Instruction Comprehension, Perception Reasoning, Generalized Planning, Affordance Prediction, and Failure Analysis. It includes 15 capabilities, 25 tasks, and more than 6,000 Q&A pairs. The dataset is curated with attention to realism, encompassing diverse robot types, object properties, and environments. The paper also proposes a planning evaluation framework that uses world-simulation rollouts with DAGs to assess embodied feasibility.

**Strengths:**

1. Comprehensive Scope and Structure: RoboBench uniquely combines five major cognitive dimensions, directly tracing the manipulation pipeline from instruction to error recovery, surpassing existing embodied benchmarks in breadth and integration.
2. Novel Planning Evaluation Framework: The MLLM-as-world-simulator approach moves beyond text matching by assessing step-by-step plan execution feasibility using ground-truth action lists and annotated DAGs.
3. The paper is easy to follow and well-strutured.

**Weaknesses:**

1. While the empirical evaluation is rigorous, the paper lacks a deeper theoretical analysis of cognitive failure patterns, limitations in MLLMs’ reasoning processes, or broader learning-theoretic implications. Theoretical insights or analyses, such as attention-based analysis, probing, or architectural examination, are limited.
2. Limited Analysis of Model Differences: The results show performance gaps but do not explore how different model designs, training data, or prompt styles affect them. For instance, is the weakness caused by weak visual-language connection or missing training examples? Some focused tests or discussion of these factors would make the analysis more insightful than just reporting performance numbers.
3. It would be helpful if the author could summarize the key insights gained from this benchmark and its evaluation results, particularly regarding how these findings could guide future improvements in MLLMs.

**Questions:**

See weakness section.

---

### Meta-Review · Area_Chair_cxPy · 2026-01-08

**Summary:**

The major concern of reviewers is that the work focuses on more engineering-oriented than research-focused. Much of the data is repurposed from existing benchmarks without a notable principled curation method, which limits the conceptual contribution. The paper is not ready to be accepted at the current form.

**Reviewer Concerns:**

Reviewers raised major concerns about the MLLM-as-world-simulator framework, noting that it effectively functions as an LLM-as-judge. As such, it may inherit well-known evaluation biases, such as favoring models with similar architectures or response styles, and the paper does not currently discuss any strategies to mitigate these issues. One reviewer also acknowledged the rigor of the empirical results but felt the paper falls short in explaining why certain cognitive failure patterns emerge or in discussing the broader learning-theoretic implications. In addition, the use of synthetically perturbed plans for failure analysis was viewed as a limitation, with reviewers questioning whether these synthetic errors accurately capture the diversity and complexity of failures observed in real-world robotic execution. Also, stronger evidence will be useful to justfiy of the reliability of the human-annotated DAGs, such as inter-annotator agreement, and for clearer justification of why this benchmark is needed rather than simply combining existing benchmarks.

**Reviewer Scores:**

There is no rebuttals and thus no reviewers changed their scores.

---

### Decision · Program_Chairs · 2026-01-26

Reject